# Neurofilaments in spinocerebellar ataxia type 3: blood biomarkers at the preataxic and ataxic stage in humans and mice

Carlo Wilke[1,2] (iD), Eva Haas[3,4], Kathrin Reetz[5,6], Jennifer Faber[7,8], Hector Garcia-Moreno[9,10], Magda M Santana[11], Bart van de Warrenburg[12], Holger Hengel[1,2], Manuela Lima[13], Alessandro Filla[14], Alexandra Durr[15], Bela Melegh[16], Marcella Masciullo[17], Jon Infante[18], Paola Giunti[9,10], Manuela Neumann[2,19], Jeroen de Vries[20], Luis Pereira de Almeida[11] (iD), Maria Rakowicz[21], Heike Jacobi[8,22], Rebecca Schüle[1,2], Stephan A Kaeser[1,2], Jens Kuhle[23], Thomas Klockgether[7,8], Ludger Schöls[1,2], SCA3 neurofilament study group[‡], Christian Barro[23,†], Jeannette Hübener-Schmid[3,4,†] & Matthis Synofzik[1,2,†,*] (iD)

## Abstract

With molecular treatments coming into reach for spinocerebellar ataxia type 3 (SCA3), easily accessible, cross-species validated biomarkers for human and preclinical trials are warranted, particularly for the preataxic disease stage. We assessed serum levels of neurofilament light (NfL) and phosphorylated neurofilament heavy (pNfH) in ataxic and preataxic subjects of two independent multicentric SCA3 cohorts and in a SCA3 knock-in mouse model. Ataxic SCA3 subjects showed increased levels of both NfL and pNfH. In preataxic subjects, NfL levels increased with proximity to the individual expected onset of ataxia, with significant NfL elevations already 7.5 years before onset. Cross-sectional NfL levels correlated with both disease severity and longitudinal disease progression. Blood NfL and pNfH increases in human SCA3 were each paralleled by similar changes in SCA3 knock-in mice, here also starting already at the presymptomatic stage, closely following ataxin-3 aggregation and preceding Purkinje cell loss in the brain. Blood neurofilaments, particularly NfL, might thus provide easily accessible, cross-species validated biomarkers in both ataxic and preataxic SCA3, associated with earliest neuropathological changes, and serve as progression, proximity-to-onset and, potentially, treatment-response markers in both human and preclinical SCA3 trials.

1 Hertie Institute for Clinical Brain Research (HIH), Center of Neurology, University of Tübingen, Tübingen, Germany
2 German Center for Neurodegenerative Diseases (DZNE), University of Tübingen, Tübingen, Germany
3 Institute of Medical Genetics and Applied Genomics, University of Tübingen, Tübingen, Germany
4 Centre for Rare Diseases, University of Tübingen, Tübingen, Germany
5 Department of Neurology, RWTH Aachen University, Aachen, Germany
6 JARA-BRAIN Institute Molecular Neuroscience and Neuroimaging, Forschungszentrum Jülich, RWTH Aachen University, Aachen, Germany
7 Department of Neurology, University Hospital Bonn, Bonn, Germany
8 German Center for Neurodegenerative Diseases (DZNE), Bonn, Germany
9 Ataxia Centre, Department of Clinical and Movement Neurosciences, UCL Queen Square Institute of Neurology, London, UK
10 National Hospital for Neurology and Neurosurgery, University College London Hospitals NHS Foundation Trust, London, UK
11 Center for Neuroscience and Cell Biology, University of Coimbra, Coimbra, Portugal
12 Donders Institute for Brain, Cognition, and Behaviour, Department of Neurology, Radboud University Medical Center, Nijmegen, The Netherlands
13 Faculdade de Ciências e Tecnologia, Universidade dos Açores, Ponta Delgada, Portugal
14 Department of Neuroscience, and Reproductive and Odontostomatological Sciences, Federico II University Naples, Naples, Italy
15 Sorbonne Université, Institut du Cerveau et de la Moelle épinière (ICM), AP-HP, Inserm, CNRS, University Hospital Pitié-Salpêtrière, Paris, France
16 Department of Medical Genetics, Szentagothai Research Center, University of Pécs Medical School, Pécs, Hungary
17 Spinal Rehabilitation Lab (SPIRE), IRCCS Fondazione Santa Lucia, Rome, Italy
18 Service of Neurology, University Hospital Marqués de Valdecilla (IDIVAL), University of Cantabria (UC), Centro de Investigación Biomédica en Red de Enfermedades Neurodegenerativas (CIBERNED), Santander, Spain
19 Department of Neuropathology, University of Tübingen, Tübingen, Germany
20 Department of Neurology, University Medical Centre Groningen, University of Groningen, Groningen, The Netherlands
21 First Department of Neurology, Institute of Psychiatry and Neurology, Warsaw, Poland
22 Department of Neurology, University Hospital of Heidelberg, Heidelberg, Germany
23 Neurology, Departments of Medicine, Biomedicine and Clinical Research, University Hospital Basel, University of Basel, Basel, Switzerland
*Corresponding author. Tel: +49 7071 2982060; E-mail: matthis.synofzik@uni-tuebingen.de
†These authors contributed equally to this work as last authors
‡See Appendix 1

**Keywords** knock-in mouse model; neurofilament light chain; phosphorylated neurofilament heavy chain; presymptomatic stage; spinocerebellar ataxia type 3

**Subject Categories** Biomarkers; Neuroscience

## Introduction

Spinocerebellar ataxia type 3 (SCA3), also known as Machado–Joseph disease, is the most common dominantly inherited form of degenerative ataxia, caused by an expanded CAG repeat in the *ATXN3* gene and marked by irreversible decline in motor function already in mid-life (Costa Mdo & Paulson, 2012; Rub *et al*, 2013). Advances in the understanding of the toxic gain-of-function mechanisms underlying SCA3 neurodegeneration have opened a window for targeted molecular therapies (Paulson *et al*, 2017; Ramani *et al*, 2017). Particularly, interventions with antisense oligonucleotides (ASOs) targeting mutated *ATXN3* show promising results in mitigating the molecular, pathological and behavioural disease-associated changes in a SCA3 mouse model (McLoughlin *et al*, 2018). ASO treatments might allow preventing the neurodegenerative process even before the occurrence of clinical symptoms (Finkel *et al*, 2017; Winter *et al*, 2019). However, to pave the way for upcoming trials of these promising therapies, easily accessible, objective and sensitive outcome parameters are urgently needed to track disease progression in both the preataxic and ataxic stage of SCA3 disease. Such parameters require validation in large human SCA3 cohorts with standardised phenotyping and in SCA3 mouse models, as mouse models allow comprehensive neuropathological validation and preclinical treatment trials, even already during the presymptomatic stage.

In this cross-species biomarker study, we propose serum concentrations of neurofilament light (NfL) and phosphorylated neurofilament heavy (pNfH) as easily accessible, objective and sensitive blood biomarkers of disease severity in SCA3. Neurofilaments (Nfs) are neuron-specific cytoskeletal proteins, with constant low-amount release from neurons in an age-related manner (Disanto *et al*, 2017; Gaetani *et al*, 2019; Khalil *et al*, 2020) and rapid increase in response to axonal damage irrespective of the underlying cause (e.g. traumatic, vascular, inflammatory, degenerative injury). It is yet unknown whether Nf elevations relate to passive release from damaged axons and/or upregulated protein production and secretion reflecting attempted axonal regeneration (Paterson *et al*, 2019). With novel ultra-sensitive single-molecule array (Simoa) assays, Nfs are now reliably quantifiable in peripheral blood (Wilke *et al*, 2016; Khalil *et al*, 2018). In fact, while the correlation between Nf levels in the extracellular compartment of the brain and the cerebrospinal fluid (CSF) is yet unknown, a close correlation between Nf levels in the CSF and the peripheral blood has been established for several neurodegenerative diseases (Bacioglu *et al*, 2016; Wilke *et al*, 2016, 2019; Khalil *et al*, 2018; Gaetani *et al*, 2019), including also SCA3 (Li *et al*, 2019). Hereby, Nf levels in blood are roughly 2.5% of the levels in the CSF (Disanto *et al*, 2017).

Our previous work in a mixed cohort of repeat-expansion spinocerebellar ataxias (SCAs) indicated that blood concentrations of NfL in multisystemic repeat SCAs are increased at the ataxic disease stage (Wilke *et al*, 2018), and this also been reported specifically for SCA3 (Li *et al*, 2019). However, all these previous studies were confined to single-centre, single-assay and human-only assessments without any neuropathology associations, as well as to cross-sectional data. Moreover, they were all restricted to NfL only, while pNfH might also allow capturing neuronal disintegration and particularly axonal damage in neurodegenerative disease, possibly capturing differential features of the neurodegenerative process compared to NfL (Khalil *et al*, 2018; Wilke *et al*, 2019).

We here hypothesised that both serum Nfs might serve as blood biomarkers of disease severity in both human SCA3 and mouse models, expecting increased concentrations at both the ataxic and preataxic stage, with increases in preataxic subjects occurring particularly in proximity to the onset of ataxia. We measured serum concentrations of both Nfs in cross-sectional samples of ataxic and preataxic SCA3 subjects and controls in two independent multicentric cohorts, using two independent ultra-sensitive single-molecule array (Simoa) approaches for each of both Nfs, and correlated Nf levels with measures of disease severity. We expected the blood Nf increases in human SCA3 to be paralleled by blood Nf increases in SCA3 animal models, also starting already in the presymptomatic stage and at the earliest stages of SCA3 neurodegeneration. We therefore assessed plasma NfL and pNfH also in a SCA3 knock-in mouse model (preprint: Haas *et al*, 2020; Martier *et al*, 2019) across presymptomatic and symptomatic disease stages, correlating plasma concentrations of both Nfs with the temporal course of phenotypic and neuropathological disease features, including brain ataxin-3 protein levels and aggregation.

## Results

### Serum NfL levels are increased at the ataxic stage of SCA3

In cohort #1 (Fig 1A), serum concentrations of NfL were significantly higher in ataxic SCA3 subjects (34.8 pg/ml (28.3–47.0), median and IQR) than in controls (8.6 pg/ml [5.7–11.7]; $U = 151$, $z = 10.1$, $P < 0.001$, $r = 0.82$). In cohort #2 (Fig 1B), NfL levels were also significantly higher in ataxic SCA3 subjects (85.5 pg/ml [70.2–100.2]) than in controls (19.4 pg/ml [15.1–25.4]; $U = 16$, $z = 6.98$, $P < 0.001$, $r = 0.81$). This confirmed the NfL increase in a second, independent cohort with an independent immunoassay. NfL levels differentiated between ataxic SCA3 subjects and controls with high accuracy (cohort #1: AUC = 0.97 (0.95–1.00), $P < 0.001$, optimal cut-off: 20.0 pg/ml, 98.7% sensitivity, 92.2% specificity; cohort #2: AUC = 0.99 (0.97–1.00), $P < 0.001$, optimal cut-off: 50.9 pg/ml, 92.6% sensitivity, 100% specificity). If corrected for age, the NfL increase in ataxic SCA3 subjects remained highly significant in both cohort #1 ($F_{(1,147)} = 406.54$, $P < 0.001$; based on a linear model with the factors group, age and their interaction, $R^2 = 0.82$; Fig 1C) and cohort #2 ($F_{(1,70)} = 169.49$, $P < 0.001$, $R^2 = 0.79$; Fig 1D).

### Serum NfL levels are increased at the preataxic stage of SCA3

In cohort #1 (Fig 1A), NfL levels of preataxic SCA3 subjects (29.1 pg/ml [15.9–43.7]) were significantly higher than in controls ($U = 72$, $z = 3.55$, $P < 0.001$, $r = 0.39$) and did not differ significantly from those of ataxic SCA3 subjects ($U = 204$, $z = 1.48$,

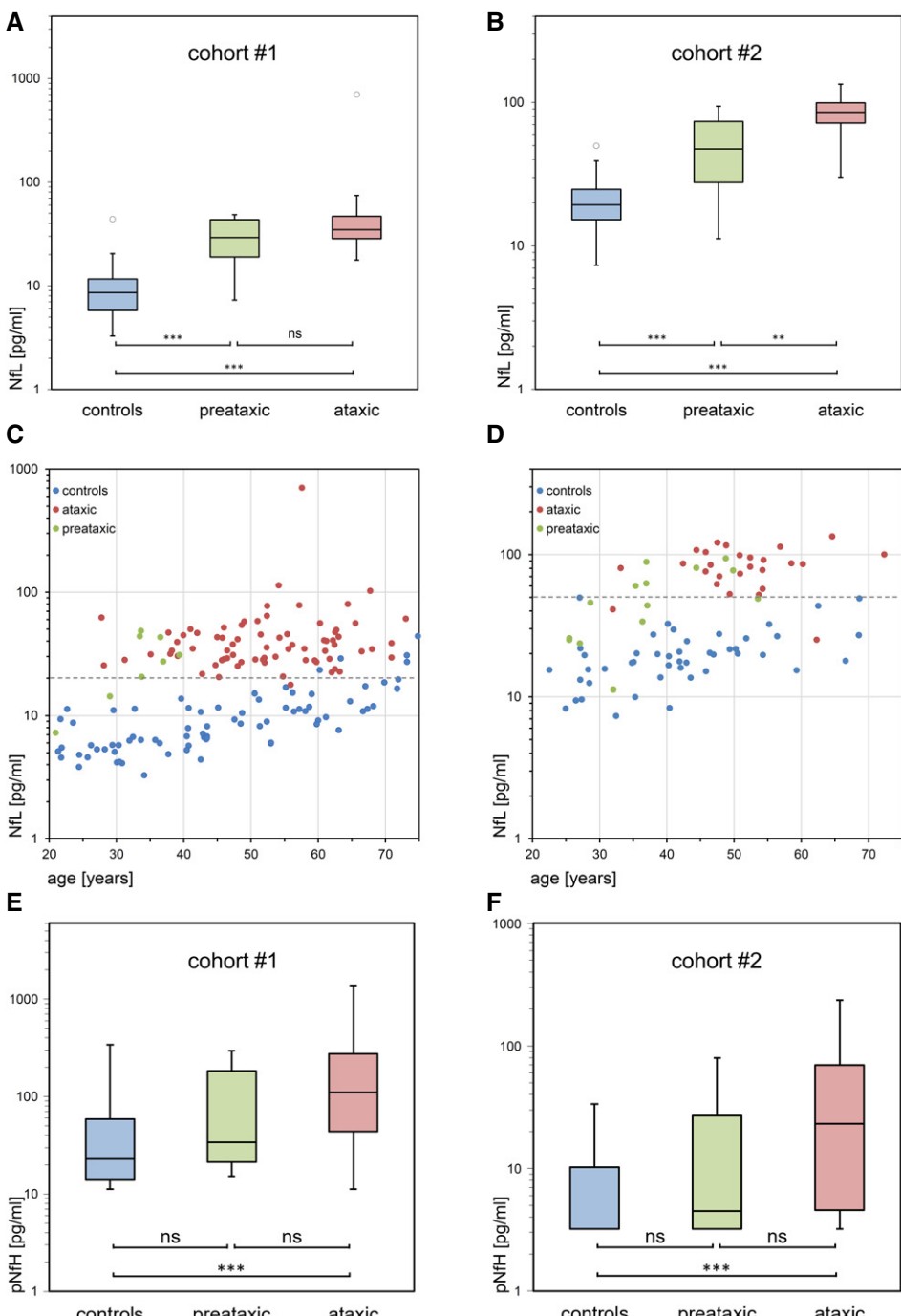

**Figure 1. Serum NfL and pNfH concentrations in the preataxic and ataxic stage of SCA3.**

A–D  Serum NfL concentrations of preataxic (green) and ataxic (red) SCA3 subjects and controls (blue) were measured in two independent cohorts, each with a different Simoa approach: cohort #1, recruited by the ESMI consortium (A, C), and cohort #2, recruited by the EuroSCA/RiSCA consortium (B, D). Boxes show the ranges between lower and upper quartiles, the central bands show the medians, and the whiskers show data within 1.5•IQR of the median, with dots representing outliers. Groups were compared with Mann–Whitney *U*-tests (***$P < 0.001$, **$P < 0.01$, ns $P \geq 0.05$, two-tailed, Bonferroni-corrected; see Appendix Table S3 for detailed statistics). In the scatter plots, the individual NfL values were plotted as a function of subjects' age. The dashed grey lines visualise the optimal cut-offs for differentiating ataxic SCA3 subjects from controls in each cohort (cohort #1: 20.0 pg/ml, 98.7% sensitivity, 92.2% specificity; cohort #2: 50.9 pg/ml, 92.6% sensitivity, 100% specificity; cut-offs were derived by maximising Youden's index irrespective of age). Note the logarithmic scale of the *y*-axes.

E, F  Serum pNfH levels of preataxic and ataxic SCA3 subjects and controls were also measured in both cohorts, each with a different Simoa approach (two-tailed Mann–Whitney *U*-tests, Bonferroni-corrected; see Appendix Table S3 for detailed statistics).

Source data are available online for this figure.

$P = 0.143$, $r = 0.16$, Bonferroni-corrected for multiple comparisons, respectively). In cohort #2 (Fig 1B), NfL levels of preataxic SCA3 subjects (47.3 pg/ml [25.5–78.0]) were also significantly increased ($U = 88$, $z = 4.18$, $P < 0.001$, $r = 0.53$), yet significantly lower than in ataxic SCA3 subjects ($U = 74$, $z = 3.16$, $P = 0.001$, $r = 0.49$, Bonferroni-corrected). Within preataxic subjects, 75% (cohort #1) and 43% (cohort #2) of NfL levels were above the optimal cut-off separating ataxic subjects from controls. The NfL increase in preataxic SCA3 subjects remained highly significant if corrected for age, both in cohort #1 ($F(1,81) = 99.27$, $P < 0.001$; based on a linear model with the factors group, age and their interaction, $R^2 = 0.68$, Fig 1C) and cohort #2 ($F(1,58) = 59.82$, $P < 0.001$, $R^2 = 0.56$, Fig 1D).

### Serum NfL levels reflect disease severity

Neurofilament light levels of ataxic SCA3 subjects significantly correlated with disease severity ($r = 0.43$, $P < 0.001$), as captured by the SARA score (Fig 2A). The association between NfL and disease severity remained highly significant if corrected for age ($r = 0.41$, $P < 0.001$), disease duration ($r = 0.41$, $P < 0.001$) and CAG repeat length ($r = 0.47$, $P < 0.001$) as possible confounders, as assessed by partial correlations.

### Serum NfL levels may reflect longitudinal disease progression

In a subset of ataxic SCA3 subjects ($n = 35$, all from cohort #2), prospective longitudinal SARA scores were available to estimate intraindividual disease progression, as quantified by the annual change of the SARA score. Subjects' cross-sectional NfL levels significantly correlated with the annual change of the SARA score ($r = 0.34$, $P = 0.045$), suggesting that serum NfL may also reflect longitudinal disease progression. This positive association was confirmed with a similar effect size ($r = 0.29$) also when adjusting for baseline SARA score, showing a statistical trend in this relatively small subcohort ($P = 0.100$, 2-sided test). Accordingly, subjects with high disease progression (annual SARA score increase $\geq 0.71$ points/year, $n = 18$, subset defined by median split) had significantly higher serum levels of NfL than subjects with low disease progression ($n = 17$; $P = 0.018$, $r = 0.40$; Fig 2B).

### Association of NfL levels with repeat length and age

We analysed the association of NfL levels with age and CAG repeat length in SCA3 mutation carriers with a linear model, using the pooled data of both preataxic and ataxic subjects. The highly significant predictors of the NfL level were age ($F(1,113) = 40.54$, $P < 0.001$), its square ($F(1,113) = 7.91$, $P = 0.006$) and repeat length ($F(1,113) = 22.01$, $P < 0.001$; total explained variance: $R^2 = 0.37$; Fig 2C). The model demonstrated that, for a given age, each increase in the CAG repeat count was associated with higher NfL levels. For a given CAG repeat count, the NfL level increased with age, with the steepness of the slope declining with increasing age. Thus, the NfL increase in SCA3 reached a plateau in older age. The sustained increase of NfL levels through the ataxic stage was reflected by the absence of correlation of NfL levels with disease duration (Fig EV1). In controls, the relation between log-transformed NfL levels and age was linear, indicating that any analyses

comparing NfL levels between carriers and controls would need to consider the physiological NfL age-related increase in controls.

### NfL levels increase with proximity to the estimated onset, with significant increases 7.5 years before ataxia onset

Neurofilament light levels of preataxic subjects increased significantly with proximity to the individually predicted onset of ataxia, as revealed by a linear regression using the pooled data of both cohorts ($F(1,21) = 39.68$, $P < 0.001$, $R^2 = 0.64$; slope: 2.89 [1.94–3.85], $P < 0.001$; Fig 3A). To compare preataxic SCA3 subjects with controls at the same age, we expressed the measured NfL level of SCA3 subjects as NfL $z$-score in relation to the age-dependent NfL distribution in controls (Fig 3B), and analysed the NfL $z$-score as a function of the time to the estimated onset of ataxia (Fig 3C). The NfL $z$-score significantly increased with preataxic subjects approaching the expected onset of ataxia ($F(1,21) = 30.78$, $P < 0.001$, $R^2 = 0.58$; slope: 0.32 [0.20–0.44], $P < 0.001$; Fig 3C), without overlap of the 95% confidence interval of controls (i.e. if $z$-score > 1.96) already 7.5 years before the expected onset (Fig 3C).

### Using serum NfL to predict time to estimated onset of ataxia

The estimated time to onset might be predicted from individual NfL measurements (for $z$-scores in the range of 1–5) using the regression depicted in Fig 3C. Moreover, the NfL $z$-score might allow delineating a preconversion stage, i.e. stratifying individual preataxic carriers close to expected symptom onset. The NfL $z$-score differentiated subjects at the late preataxic stage (i.e. carriers within 10 years of expected onset) from subjects at the early preataxic stage (i.e. more than 10 years before the expected onset) with high accuracy (AUC = 0.89 [0.76–1.00] [95% CI], $P = 0.002$). Specifically, a cut-off for the NfL $z$-score at 3.2 differentiated early from late preataxic subjects with 85% sensitivity and 90% specificity.

### Increased phosphorylated neurofilament heavy (pNfH) levels in the ataxic disease stage of SCA3

In cohort #1 (Fig 1E), SCA3 subjects at the ataxic stage had significantly higher serum pNfH concentrations (110.2 pg/ml [42.9–277.9]) than controls (22.8 pg/ml [13.9–59.1]; $U = 1,064$, $z = 6.72$, $P < 0.001$, $r = 0.55$, Bonferroni-corrected), while pNfH concentrations in preataxic SCA3 subjects (33.9 pg/ml [18.0–184.8]) were not significantly increased compared to controls ($U = 201$, $z = 1.61$, $P = 0.109$, $r = 0.17$). Likewise, in cohort #2 (Fig 1F), serum pNfH levels were significantly higher in ataxic SCA3 subjects (23.2 pg/ml [3.2–71.2]) than in controls (3.2 pg/ml [3.2–10.4]; $U = 260$, $z = 4.57$, $P < 0.001$, $r = 0.53$), while pNfH concentrations in preataxic SCA3 subjects (4.5 pg/ml [3.2–34.6]) were not significantly increased ($U = 250$, $z = 1.63$, $P = 0.103$, $r = 0.21$). These findings from two independent cohorts and assays validate pNfH increases in the ataxic stage of SCA3. Such increases might not necessarily be observed at the preataxic stage of SCA3. However, in both cohorts, pNfH levels at the preataxic stage were not significantly lower than in the ataxic stage of SCA3. While levels of NfL and pNfH were moderately correlated in controls (cohort #1: $\varrho = 0.28$, $P = 0.013$, cohort #2: $\varrho = 0.39$, $P = 0.006$), this correlation was only partially maintained in SCA3 mutation carriers (cohort #1, $\varrho = 0.13$,

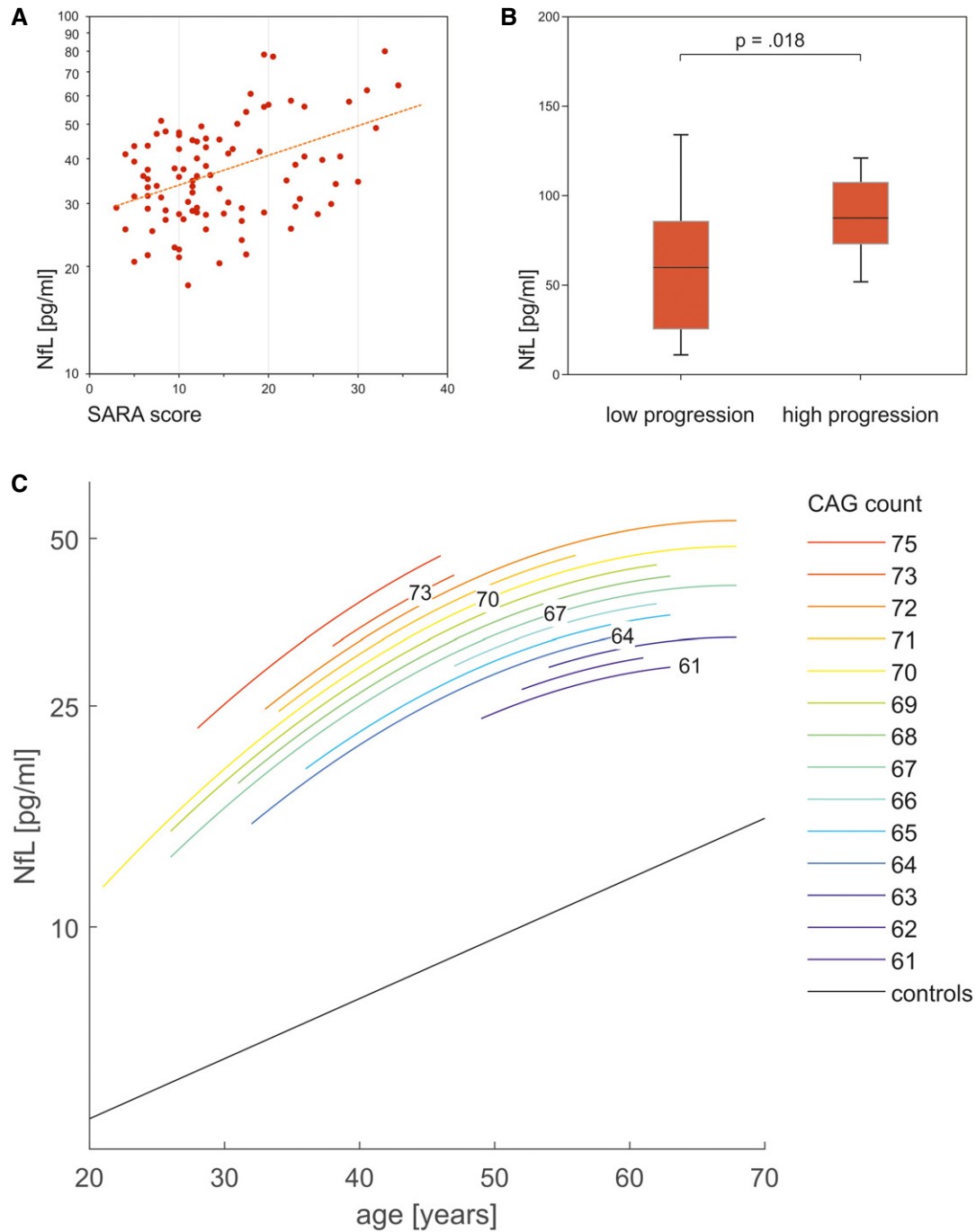

**Figure 2. Associations of NfL levels with disease severity, disease progression, age and CAG repeat length in SCA3.**

A  Serum NfL levels of ataxic SCA3 subjects significantly correlated with disease severity, as quantified by the Scale for the Rating and Assessment of Ataxia (SARA) score ($r$ = 0.43, $P < 0.001$, Pearson's correlation, two-tailed test).

B  The cross-sectional NfL levels also reflected longitudinal disease progression, as quantified by the annual SARA score change (available for 35 subjects). Boxes show the ranges between lower and upper quartiles, the central bands show the medians, and the whiskers show data within 1.5•IQR of the median, with dots representing outliers. Subjects with high disease progression (annual SARA score increase ≥ 0.71 points/year, $n$ = 18, median split) had significantly higher serum levels of NfL than subjects with low disease progression ($n$ = 17; $P$ = 0.018, $r$ = 0.40, Mann–Whitney $U$-test, two-tailed; see Appendix Table S3 for detailed statistics).

C  We modelled serum NfL levels (log-transformed) in SCA3 carriers ($n$ = 123) with the predictors age and *ATXN3* CAG repeat length, their squares and all possible interactions (for details, see Results and Materials and Methods). The highly significant predictors (age, its square and repeat length, and the intercept, all $P < 0.001$, explained variance: $R^2$ = 0.37) were used to generate the diagram. For a given age, each increase in CAG repeat count was associated with higher NfL concentrations. The steepness of the slopes declined with increasing age. In controls (black, $n$ = 125), the relation between NfL level (log-transformed) and age was linear.

Source data are available online for this figure.

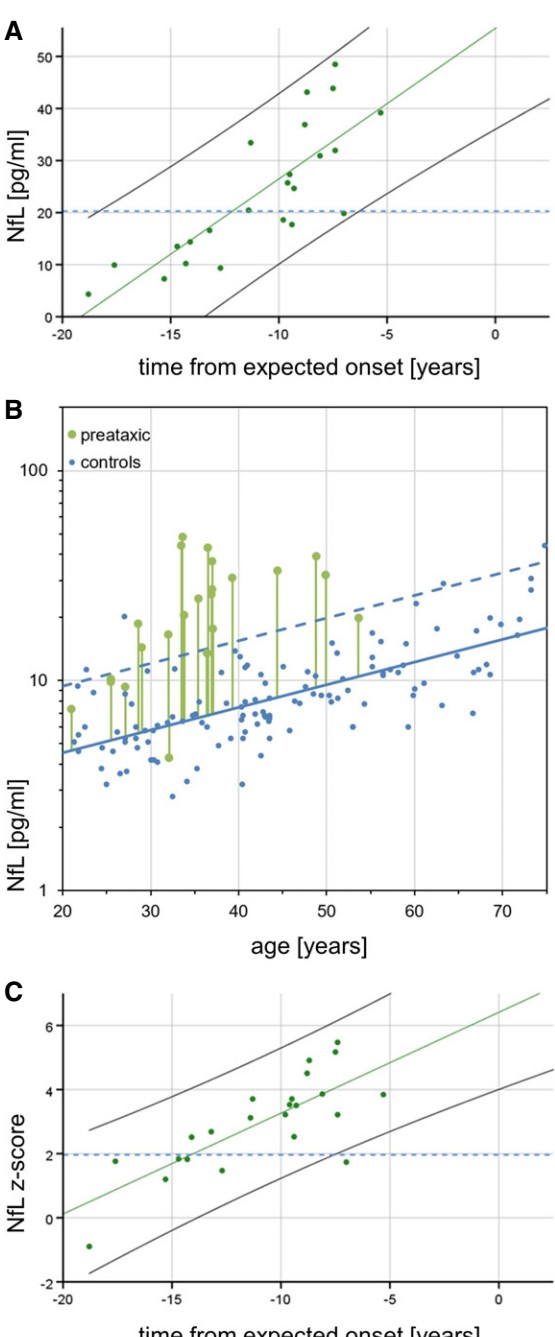

**Figure 3. Association between serum NfL and time from expected onset in preataxic SCA3.**

A  Serum NfL levels in preataxic SCA3 subjects (cross-sectional data) were plotted over the time from the individually estimated ataxia onset. NfL levels increased significantly with proximity to the estimated onset ($F(1,21) = 39.68$, $P < 0.001$, $R^2 = 0.64$). For benchmarking the levels of preataxic subjects, the blue dashed line visualises the optimal cut-off for differentiating ataxic SCA3 subjects from controls in the pooled cohort (20.3 pg/ml, 99.0% sensitivity, 95.2% specificity).

B  To determine the time at which NfL levels become significantly increased in the preataxic stage of SCA3, NfL levels of preataxic carriers need to be related to NfL levels of controls *at the same age*, as NfL levels physiologically increase with age. For each preataxic subject, the difference between the measured NfL value (green dot) and the NfL value predicted for control subjects of the same age (solid blue line) was visualised by the length of the vertical green line. Standardisation of this difference relative to the NfL distribution in controls (95% CI of the data: dashed blue line) yielded the individual NfL z-score, which was plotted over each subject's estimated time to onset in the next panel.

C  In preataxic SCA3 subjects, NfL z-scores increased significantly ($F(1,21) = 30.78$, $P < 0.001$, $R^2 = 0.58$) with subjects approaching the expected age of onset. NfL levels of preataxic subjects were significantly increased compared to controls (i.e. z-score > 1.96) 7.5 years before the expected onset, indicated by the non-overlapping 95% confidence intervals of SCA3 subjects (black solid line) and controls (blue dashed line).

Source data are available online for this figure.

$P = 0.240$, cohort #2: $\varrho = 0.33$, $P = 0.034$), suggesting that pNfH might reflect a partly different feature/process in SCA3 than NfL.

## Intraindividual Nf stability and sample size estimation for intervention trials using Nf levels as outcome parameter

Short-term longitudinal assessment of Nf levels over 6 weeks ($n = 21$) demonstrated high intraindividual stability for both NfL and pNfH in SCA3, as evidenced by very high intraclass correlation coefficients (ICCs; NfL: 0.983 [0.960–0.993] [95% CI], pNfH: 0.995 [0.987–0.998]; Fig 4A and B). Sample size estimates for hypothetical intervention trials using the reduction of Nf levels as outcome parameter indicate, e.g., that $\approx$ 15 subjects per arm would suffice to detect therapeutic effects with effect sizes as low as 20% (see Fig 4C also for visualisation of a range of other possible treatment effect sizes).

## Human neuropathology

Neuropathologic examination of SCA3 subject ID27925 (cohort #2) with an NfL level of 116 pg/ml (93. percentile within the ataxic subjects of cohort #2) and a pNfH level of 23.2 pg/ml (52. percentile) at age 48.5 years (i.e. 12 months before death) did not show major degeneration of corticospinal tract, primary motor cortex and Purkinje cell layer, but marked degeneration of spinocerebellar tract, dentate nucleus and brainstem (Appendix Table S1). Similarly, while only rare polyQ pathology was found in the primary motor cortex and Purkinje cell layer, frequent polyQ pathology was present in the dentate nucleus, brainstem, basal ganglia and anterior horn. These findings, which correspond to and further corroborate previous SCA3 neuropathology case series (Paulson, 2012; Paulson *et al*,

2017; Koeppen, 2018), preliminarily indicate that not the corticospinal tract, motor cortex and cerebellar cortex, but rather affection of the spinocerebellar tract and brainstem, might underlie the Nf increase in SCA3.

## Plasma neurofilament levels in a SCA3 mouse model are increased in proximity to phenotypic onset

To elucidate the temporal cascade of Nf concentrations in SCA3, we analysed blood concentrations of Nfs in a 304Q SCA3 knock-in mouse model with a phenotypic onset at around 8 months of age, sampling animals at age 2 months (= early presymptomatic), 6 months (= late presymptomatic), 12 months (= early symptomatic) and > 18 months (= late symptomatic disease stage). While plasma concentrations of NfL were not yet increased at 2 months, they significantly increased already at age 6 months ($P < 0.001$) and 12 months ($P < 0.001$, Bonferroni-corrected) in heterozygous mutants compared to wild-type animals, with this effect levelling out in late symptomatic mice at age > 18 months (Fig 5A). Analogously, plasma concentrations of pNfH were significantly increased at 6 months ($P < 0.01$) and 12 months ($P < 0.001$) in heterozygous mutants compared to wild-type animals, but not at 2 months and > 18 months, supporting an increase in Nf blood concentrations in proximity to the phenotypic onset (Fig 5B). Exploratory analysis of Nf plasma levels in homozygous mutants confirmed the temporal cascade observed in heterozygous animals (Fig EV2). Tissue concentrations of NfL and pNfH in the cerebellum and frontal lobe did not differ between mutant and wild-type animals at the presymptomatic and symptomatic disease stage, preliminarily suggesting that the differences observed in the blood concentrations might relate not to changes of the amount of Nfs

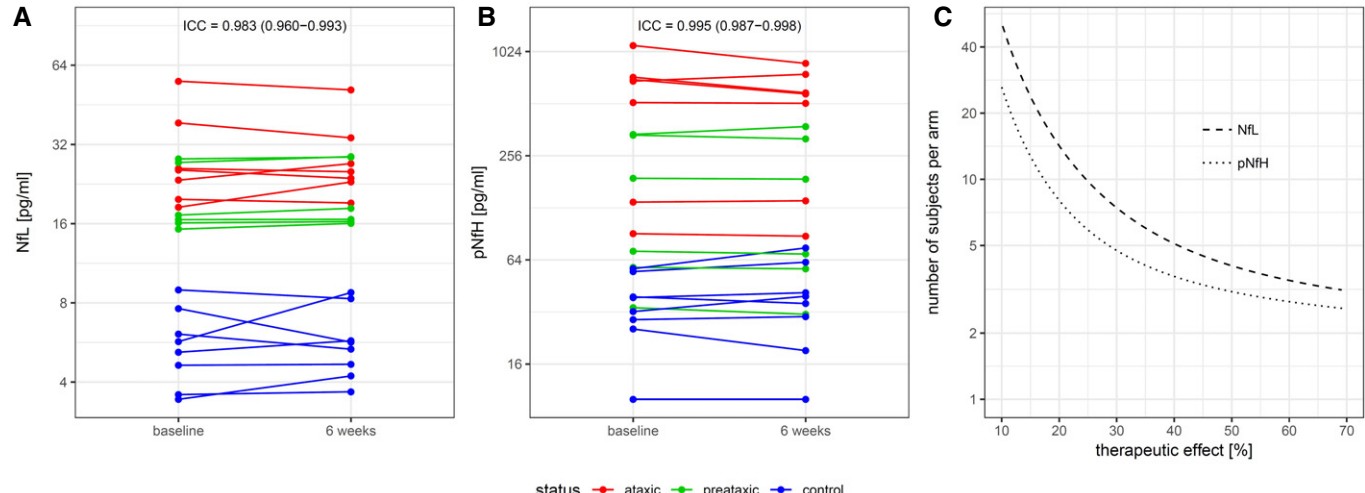

**Figure 4. Within-subject stability of Nf levels and sample size estimates for clinical trials.**

A, B   Longitudinal intraindividual stability of NfL (A) and pNfH (B) serum levels over 6 weeks, assessed in both ataxic (red) and preataxic (green) SCA3 and control (blue) subjects ($n = 21$). Data of the same individuals are linked by lines. ICC, intraclass correlation coefficient (estimate and 95% CI).

C   Sample size estimations were performed for future intervention trials using the reduction of Nf levels towards control levels as outcome measure. The estimated number of subjects per study arm is plotted over the assumed therapeutic effect for lowering the Nf level in SCA3 subjects to levels observed in healthy controls (NfL: dashed line, pNfH: dotted line).

Source data are available online for this figure.

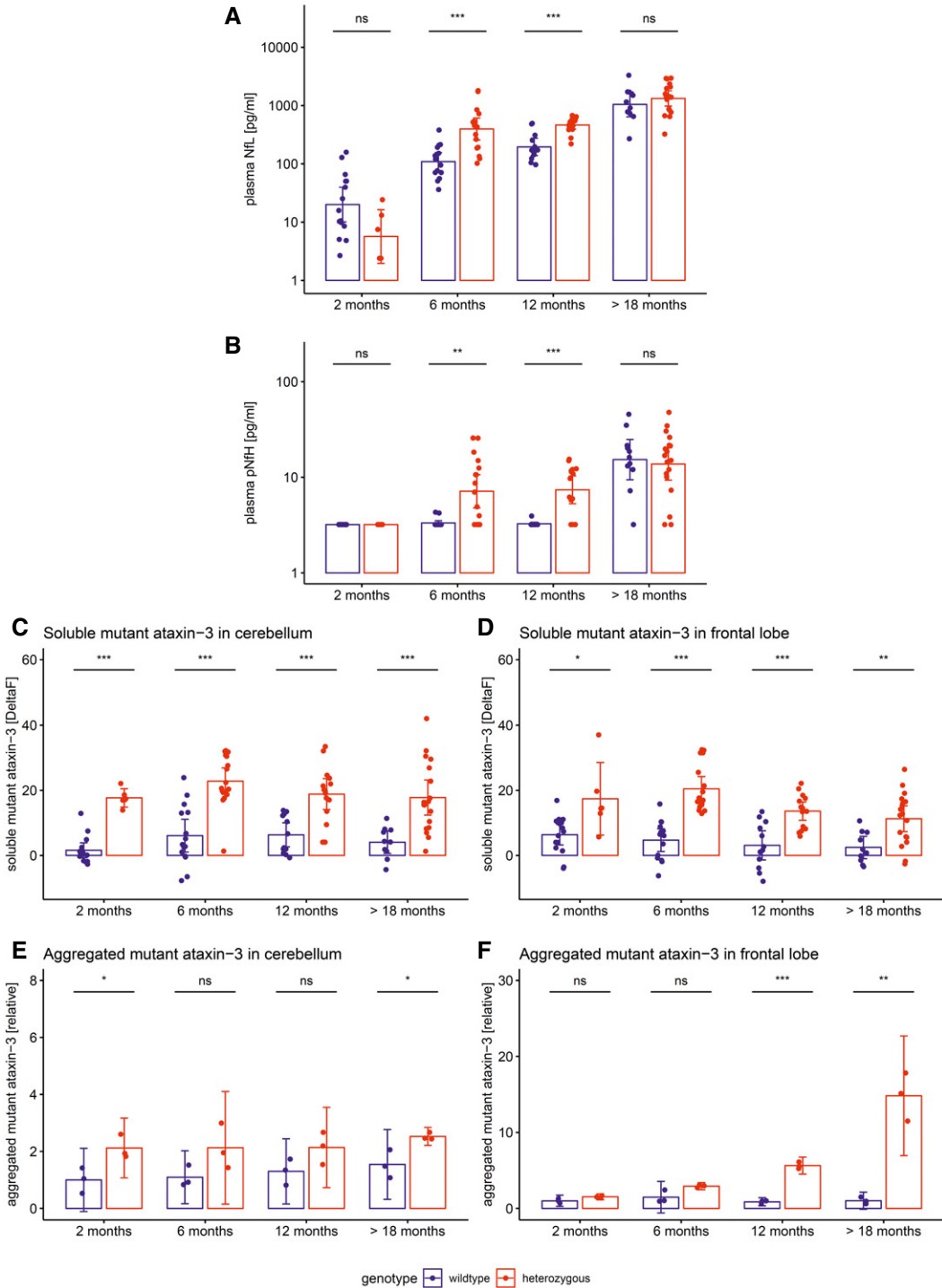

**Figure 5. Neurofilament plasma levels and brain tissue levels of soluble and aggregated ataxin-3 in the 304Q SCA3 mouse model.**

A, B  Plasma concentrations of NfL (A) and pNfH (B) were measured in the 304Q knock-in SCA3 mouse model. Heterozygous animals become symptomatic at
≈ 8 months of age.

C–F  Tissue levels of soluble and aggregated mutant ataxin-3 were measured in cerebellum (C, E) and frontal lobe (D, F). Exploratory analyses in homozygous animals
confirmed the ataxin-3 increases observed in heterozygous animals (Fig EV4).

Data information: Heterozygous and wild-type animals were compared by two-tailed unpaired *t*-tests, adjusted for unequal variances (***$P < 0.001$, **$P < 0.01$,
*$P < 0.05$, ns $P \geq 0.05$, Bonferroni-corrected; see Appendix Table S3 for detailed statistics). Dots show individual measurements, bars indicate mean ± SD (calculated for
log-transformed values).

Source data are available online for this figure.

within the brain tissue, but rather to the *release* of Nfs from the brain tissue with neurodegenerative decay (Fig EV3). Brain Nf concentrations were hereby higher in the cerebellum than in the frontal lobe, suggesting higher share of axonal tissue in the cerebellum, and thus adding further support that *axonal* damage markers —such as Nfs—might be promising markers for degenerative cerebellar disease.

Both NfL and pNfH increases in SCA3 relative to wild-type mice started earlier (age: 6 months, both males and females; Fig EV2) than the weight phenotype (age: 7 months, males; age: 12 months, females; Appendix Fig S1) and the motor phenotype (age: 18 months, both males and females; Appendix Fig S2), highlighting their value as proximity blood biomarkers for the preconversion stage also in mice. While Nf levels increased later than both aggregated and soluble cerebellar mutant ataxin-3 (age: 2 months; Fig 5C and E), ataxin-3 levels did not change throughout the disease course and in particular not with proximity to conversion (Fig 5C and E; for ataxin-3 tissue levels in homozygous mice, see Fig EV4). This indicates that, in contrast to Nfs, ataxin-3 levels might serve not as preconversion markers, but possibly rather as target-engagement markers. Taken together, our biomarkers and phenotype parameters

allow mapping a preliminary multimodal chart of disease evolution in the 304Q SCA3 mouse model from the early preataxic to the late ataxic stages of SCA3 disease (Fig 6).

### Temporal sequence of associated murine neuropathology

To elucidate the brain pathology changes temporally associated with the onset of the plasma Nf increase, we investigated the temporal sequence of neuropathology changes in SCA3 mice. Ataxin-3 aggregation started at months 2–3 and was clearly present at month 6 in regions typically affected by SCA3, particularly the cerebellum, pons and frontal cortex (Fig 7A), thus partly *preceding*, partly paralleling the onset of the plasma Nf increase. Purkinje cell (PC) loss started at months 2–3, but did not become significant before 12 months of age (Fig 7B, Appendix Fig S3), thus partly paralleling, but largely *succeeding* the onset of the Nf increase. Interestingly, at the onset of the Nf increase in blood (i.e. month 6), structural changes in PCs were microscopically more prominent than absolute PC loss (Appendix Fig S3), which might suggest that incipient structural alteration, and not only PC cell death, might contribute to Nf release in SCA3. These PC alterations were accompanied by structural

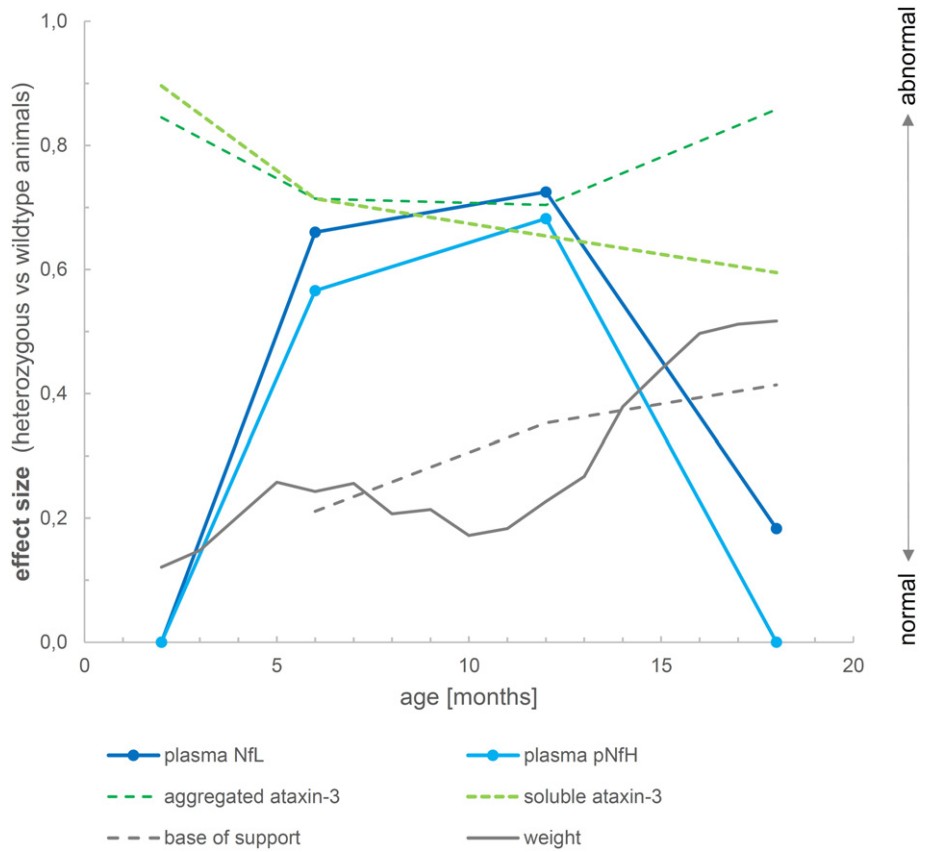

**Figure 6. Biomarker cascade of NfL, pNfH, cerebellar aggregated and soluble ataxin-3 and phenotype in 304Q knock-in SCA3 mice.**

The figure shows the degree to which each variable differs between heterozygous SCA3 mice and wild-type controls for a given age. For each variable, we quantified the deviation of heterozygous mice from controls of the same age by calculating the effect size *r* of the group comparison (independent *t*-tests). An effect size of 0 indicates that the value in heterozygous mice is not different from controls, while an effect size of 1 would indicate strong abnormality. Note that the drop in the effect size of Nf levels from 12 to 18 months thus does not imply a putative reduction of Nf levels, but rather the loss of the effect in mutants relative to controls, as controls show strong age-related Nf increases at 18 months of age. Negative effect size values were set to 0.

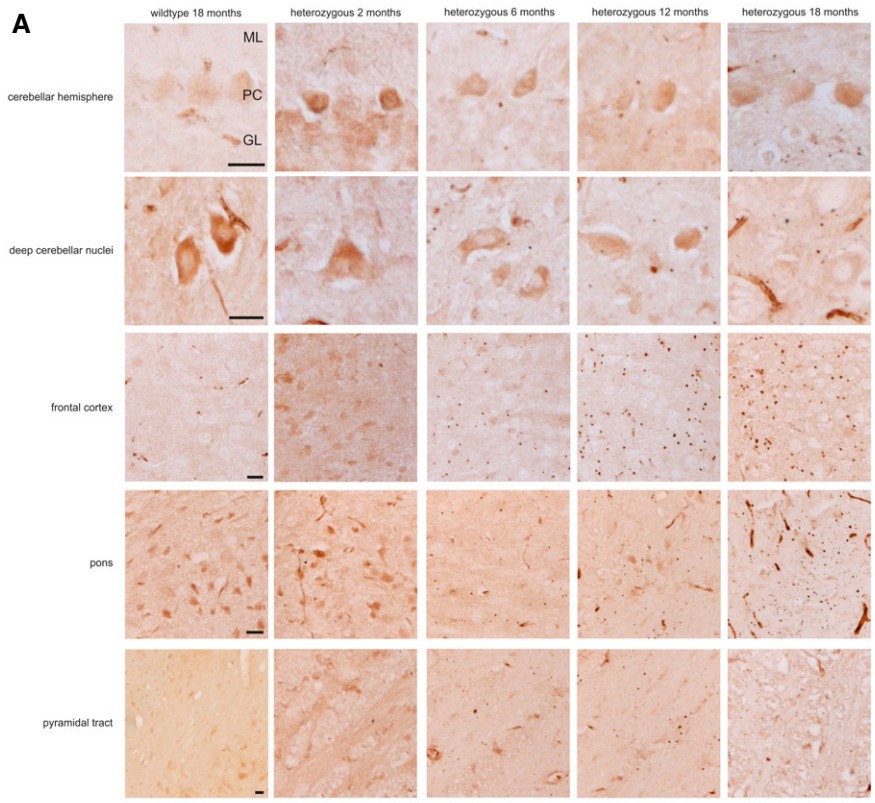

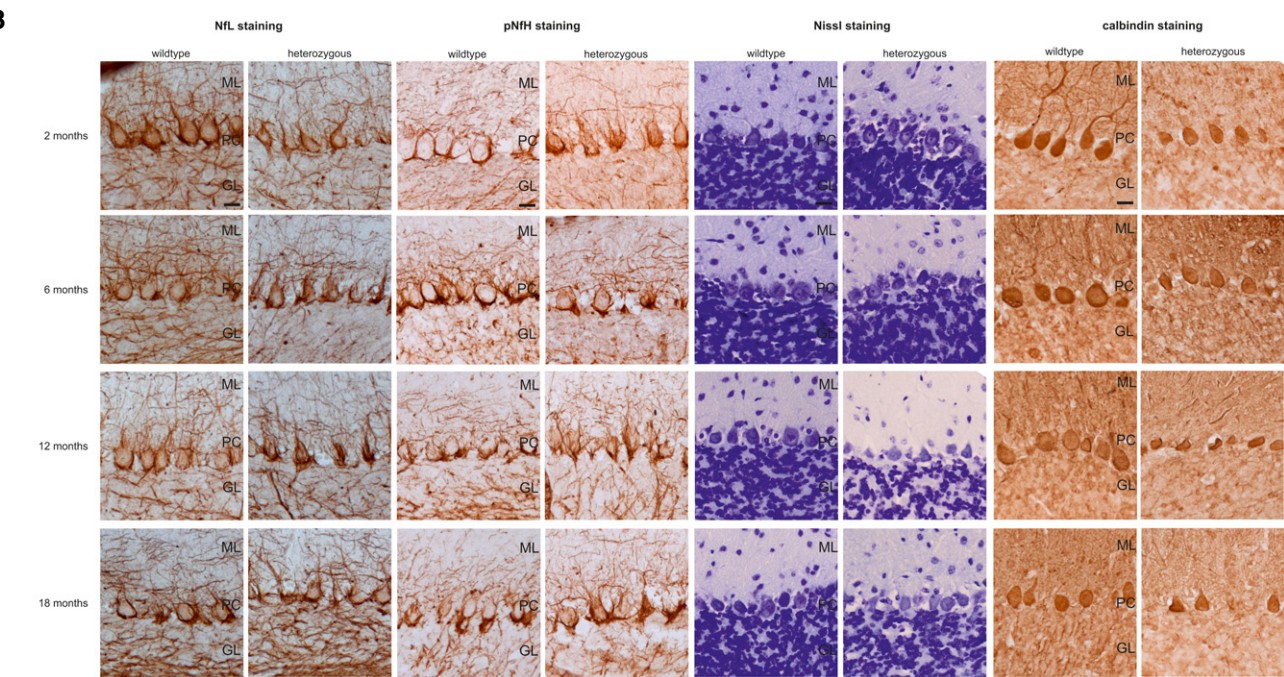

**Figure 7. Ataxin-3 immunohistochemistry and histological assessment of Purkinje cells in the 304Q SCA3 mouse model.**

A We assessed samples of cerebellar cortex, deep cerebellar nuclei, frontal cortex, pons and pyramidal tract by immunohistochemistry with ataxin-3 staining, comparing heterozygous 304Q SCA3 mice at 2, 6, 12 and 18 months of age with wild-type animals at 18 months.

B We histologically assessed the number and integrity of Purkinje cells in the cerebellar cortex by NfL, pNfH, Nissl and calbindin staining, comparing heterozygous 304Q SCA3 mice at 2, 6, 12 and 18 months of age with wild-type animals.

Data information: GL, granular layer; ML, molecular layer, PC, Purkinje cells, scale bars: 20 μm.

disturbance of the NfL and pNfH network in the cerebellar cortex (Fig 7B). No marked overall atrophy of the deep cerebellar nuclei, pons, frontal cortex and pyramidal tract was observed (Fig EV5, Appendix Fig S4). Taken together, this indicates that Nf increases in blood occur with strong effect sizes already at the earliest stages of SCA3 disease, shortly following onset of ataxin-3 aggregation and coinciding with PC structural alterations even prior to PC loss, and are not secondary to marked overall atrophy of pertinent SCA3 brain regions.

## Discussion

To pave the way for upcoming treatment trials, easily accessible surrogate biomarkers are highly warranted in SCA3 for both human and preclinical trials. In this comprehensive cross-species study—leveraging two independent large multicentric human cohorts and a knock-in mouse model, each covering both presymptomatic and symptomatic SCA3 disease stages and including both blood and neuropathology assessments—we demonstrate that (i) NfL levels are increased not only at the ataxic, but also in the preataxic stage of SCA3 in both humans and mice, increasing with repeat length and age and with proximity to expected disease onset in humans; that (ii) pNfH levels are increased at the ataxic stage of human SCA3 and at both the preataxic and ataxic stage of murine SCA3; and that (iii) these peripheral blood changes in both NfL and pNfH closely follow ataxin-3 aggregations and coincide with early PC structural alterations in the brain even prior to marked PC cell loss.

The observed increases in human NfL and pNfH serum levels in SCA3 seem to present a highly robust finding as they were each confirmed in two independent cohorts with two different Simoa assays. NfL levels differentiated SCA3 subjects from controls with high accuracy, with effect sizes comparable to state-of-the-art magnetic resonance spectroscopy methods (Adanyeguh *et al*, 2015; Joers *et al*, 2018). These Nf increases in SCA3 might not be primarily of help as a *diagnostic biomarker*, as Nfs are increased in many neurodegenerative and non-neurodegenerative conditions, including several multisystemic ataxias (Wilke *et al*, 2018). This is, however, not primarily needed for SCA3, as assessment of the *ATXN3* repeat expansion or the mutant protein meets this biomarker purpose. Rather, for preparing upcoming treatment trials, the biomarker value of NfL levels might lie in their potential as *disease severity, progression* and *stratification biomarker,* or—in terms of the Biomarker Qualification Program of the Food and Drug Administration (FDA)—in their potential as *monitoring, prognostic, response* and/or *safety biomarker* for SCA3 drug development. We here demonstrate that NfL levels indeed correlate not only with cross-sectional disease severity, but also with the longitudinal annual change of the SARA score within individuals. These significant associations, though moderate in effect size, indicate that NfL blood levels might serve as easily accessible fluid biomarkers of disease severity in SCA3. Given that NfL levels distinguish fast vs slow disease progressors, they might also aid as stratification markers of SCA3 patients in upcoming treatment trials according to categories of disease progression. As NfL levels thus reflect disease severity and disease progression rates, but not clinical disease duration (Fig EV1), they seem to capture the functional stage and ongoing neuronal decay of SCA3 disease, rather than the mere calendrical

estimate of duration of the disease. Increased NfL levels therefore might be related to spinocerebellar tract (and possibly brainstem) degeneration in SCA3, rather than, e.g., corticospinal tract degeneration (as, e.g., in ALS (Menke *et al*, 2015; Skillback *et al*, 2014)), as suggested by our human neuropathological findings. The neuropathological association observed in our study corroborates, but at the same time specifies previous relatively broad volumetric MRI findings suggesting an association of NfL levels with cerebellar volumes and brainstem volumes (Li *et al*, 2019). Our single-subject neuropathology finding, however, requires validation in larger SCA3 cohorts with combined neuropathology and Nf assessment.

Neurofilament light levels appear to be strongly influenced, inter alia, by CAG repeat length and age, as both factors were found to present independent significant predictors of individual NfL levels. For a given age, each increase in the CAG repeat count was associated with higher NfL levels. This association between NfL levels and CAG repeat length suggests a dose–response relationship for SCA3 neurodegeneration and, together with analogous findings in Huntington's disease (Byrne *et al*, 2017), more generally for polyglutamine disorders. For a given CAG repeat length, NfL levels increased with age, with the steepness of the slope decreasing with increasing age; i.e., the NfL increase in SCA3 reached a plateau in old age.

In both human cohorts, NfL levels were already increased at the preataxic stage of the disease, hereby, the individual estimated time to onset might even be predicted from individual NfL levels. The NfL increase preceded the conversion to the ataxic stage by 7.5 years, with levels further increasing with proximity to the individually predicted onset of ataxia. NfL levels might thus allow to capture the onset of degenerative neuronal loss and the subclinical progression in the preataxic stage, possibly even on a single-subject level. They might moreover aid to stratify a window of opportunity in which future disease-modifying therapies might be most effective. If future longitudinal assessments of converters confirm this NfL increase with proximity to ataxia onset (i.e. NfL as proximity/preconversion stratification biomarker), NfL might aid the selection of preataxic candidate subjects for therapeutic intervention, as invasive treatments (like ASOs) should likely be applied neither unnecessarily early nor too late in preataxic SCA3 subjects.

Short-term longitudinal assessment of Nf levels demonstrated both NfL and pNfH blood levels to be highly stable within subjects, suggesting that intraindividual biological variation is likely a minimal source of noise in observation and intervention trials using Nf blood levels as outcome measure. Our sample size estimations for future treatment trials aiming to lower Nf blood levels in SCA3 showed that already ≈ 15 subjects per arm would suffice to detect therapeutic effects, even for therapeutic effect sizes as low as 20%. This number is considerably below the cohort sizes which would likely be required for clinical endpoints, e.g. SARA score (Jacobi *et al*, 2015). Nf levels might thus provide a means for reducing trial sample size similar to or even more than the most change-sensitive MRI parameters (Reetz *et al*, 2013; Adanyeguh *et al*, 2018).

Serum levels of pNfH in both cohorts were increased at the ataxic stage of human SCA3, but—unlike NfL levels—not the preataxic stage. Given the higher sensitivity of NfL to capture changes already at the preataxic stage of human SCA3, our findings prioritise the use of NfL over pNfH in future SCA3 treatment trials. However, given their clear increase and high intraindividual stability, both markers

might still be further explored as combined fluid biomarkers in future longitudinal SCA3 trials, as they might show differential responses and dynamics to disease-modifying treatments or capture different features underlying multifaceted neurodegeneration, as suggested for other neurodegenerative diseases (Poesen & Van Damme, 2018; De Vivo et al, 2019; Winter et al, 2019).

Increases of NfL and pNfH blood levels in our two human SCA3 cohorts were paralleled by corresponding increases in our SCA3 mouse model, validating their value as neuronal damage biomarker in SCA3 across species. In mice, not only NfL, but also pNfH, showed increases before the onset of the phenotype, namely at age 6 months, thus validating the preataxic increase of Nf blood levels observed in human SCA3. Moreover, again mirroring the findings from human SCA3, these increases were sustained in SCA3 mice during the further disease course (i.e. at age 12 months), before the effects were then levelled out by the age-related increases in Nfs in elderly wild-type mice (i.e. at age > 18 months). Taking together our human and murine results, the temporal dynamics of Nfs in SCA3 might hence be non-linear, with a boost of Nfs (in particular NfL) in proximity to phenotype conversion, and stabilisation of increased levels afterwards. Similar models of Nf dynamics have been proposed for other neurodegenerative diseases such as amyotrophic lateral sclerosis (ALS) (Weydt et al, 2016) or frontotemporal dementia (FTD) (van der Ende et al, 2019). Hence, Nf levels (in particular NfL) might be particularly sensitive to neuronal decay in the *early* disease stage of SCA3, where biomarkers for tracking disease intensity would be most needed given the absence of change-sensitive clinical parameters.

The SCA3 mouse model also allowed us to assess the associated histological and immunohistochemical change at the time of the Nf increase. The peripheral blood Nf increases occur already at the earliest stages of SCA3 disease, shortly following onset of SCA3 brain hallmarks (ataxin-3 aggregation) and coinciding with incipient SCA3 neurodegeneration (PC structural alterations) even prior to PC cell loss. This finding again highlights the power of *peripheral* Nfs to coincide with *early central nervous* SCA3 neurodegenerative processes.

Going beyond mere immunohistochemistry and histology, the mouse model also allowed us to more specifically relate the biomarker cascade of peripheral Nf levels to central ataxin-3 levels. Both soluble and aggregated mutant ataxin-3 increases in the cerebellum were detectable earlier than Nf increases (Fig 5C and E), in line with the notion that they occur further upstream in SCA3 pathogenesis compared to the axonal damage reflected by Nf levels. Interestingly, in the frontal lobe aggregated mutant ataxin-3 seemed to increase later than blood Nf levels (Fig 5D and F), again corroborating the notion that the Nf increase in blood does not lag behind largely after ataxin-3 aggregation in some brain areas. In contrast to Nfs, however, ataxin-3 levels did not rise further during the late presymptomatic and early symptomatic stage in SCA3 mice. This indicates that while mutant ataxin-3 levels might serve as target-engagement biomarker, they are of limited value as biomarker of disease severity or progression (in contrast to Nfs), at least in this SCA3 mouse model. Nfs and mutant ataxin-3 might thus be used as complementary composite biomarkers with differential utility in future preclinical SCA3 mouse treatment trials.

Taken together, our study thus not only confirms, but rather largely extends recent studies indicating an NfL increase in human

SCA3 (Wilke et al, 2018; Li et al, 2019), as these studies were non-multicentric, single-assay, human- and NfL-only studies, lacking the comparative assessment of several molecular biomarkers, the relative biomarker cascade mapping and any neuropathology associations. Specifically, our study extends previous research in multiple ways, namely by: demonstrating the association of NfL levels with longitudinal disease progression; providing short-term longitudinal Nf measurements as the basis for sample size estimations for future treatment trials; cross-validating two different Nf assays; delineating in preataxic SCA3 the increase of NfL levels with temporal proximity to the expected disease onset, thus allowing to define a preconversion stage; revealing NfL and pNfH increases also in murine SCA3, thus preparing the translational path for Nfs as outcome measures in preclinical studies; relating the Nf changes to other molecular biomarkers (such as ataxin-3), thus placing Nfs into a larger biomarker cascade landscape; and relating peripheral Nf changes to central human and murine SCA3 brain pathology, thus providing first preliminary insights into the early SCA3 brain degeneration processes associated with the blood Nf increase.

Our study has several limitations. First, longitudinal assessment of Nf levels covering not only a short-term interval (as in our study), but also longer intervals is warranted to confirm our cross-sectional Nf findings. Such long-term *intra*individual temporal dynamics of Nfs might be even more sensitive to early stages of degeneration and better suited to capture individual disease progression, as indicated by longitudinal Nf studies of genetic FTD and Alzheimer's disease (van der Ende et al, 2019; Preische et al, 2019). Second, association of Nf levels with larger neuropathological and/or MRI datasets is warranted to determine the regions of the central nervous system, which mainly drive the Nf increases, and to disentangle the differential dynamics and systems contributing to NfL vs pNfH increases. Yet, our study has identified the first set of fluid biomarker candidates for both human and preclinical treatment trials in SCA3.

# Materials and Methods

### Human cohorts

Serum Nf concentrations were assessed in cross-sectional samples from two independent multicentric international SCA3 cohorts. Cohort #1 was recruited by the ESMI consortium (European Spinocerebellar Ataxia Type 3/Machado-Joseph Disease Initiative, recruitment period: 2016–2018), comprising 83 SCA3 subjects and 77 healthy controls. Cohort #2 comprised 41 SCA3 subjects and 48 healthy controls from the EuroSCA (European integrated project on spinocerebellar ataxias) and RiSCA (prospective study of individuals at risk for spinocerebellar ataxia) cohorts (recruitment period: 2008–2016, clinicaltrials.gov: NCT01037777) (Jacobi et al, 2013, 2015). The recruitment of these studies included subjects with genetically confirmed SCA3 (*ATXN3* repeat length ≥ 53; for determination of CAG repeat length, see Appendix Supplementary Methods), their first-degree relatives (i.e. siblings and children) and unrelated neurologically healthy controls. Based on their score on the Scale for the Assessment and Rating of Ataxia (SARA) (Schmitz-Hubsch et al, 2006), SCA3 mutation carriers were classified as either ataxic (SARA score ≥ 3, cohort #1: 75 subjects, cohort #2: 27 subjects) or preataxic (SARA score < 3, cohort #1: 8 subjects, cohort #2: 14 subjects).

Controls comprised mutation-negative first-degree relatives of SCA3 carriers and unrelated healthy individuals, all without symptoms or signs of neurodegenerative disease. Sample size calculation was based on a piloting study indicating that 15 ataxic SCA3 subjects and 15 controls would suffice to detect significant differences of NfL serum levels between groups (assuming $\alpha = 0.01$, $\beta = 0.01$, equal group size, use of two-tailed non-parametric test) (Wilke et al, 2018). However, we here included all available SCA3 subjects in each cohort to assess associations of Nf levels with clinical and genetic variables. For 35 SCA3 subjects in cohort #2, longitudinal SARA scores were available. To assess intraindividual analyte stability, we additionally collected short-term longitudinal blood samples in a subset of subjects from cohort #2 who were available at a second visit 6 weeks after the baseline visit ($n = 21$ subjects: seven ataxic, six preataxic and eight control subjects; sampling interval: 42 days [34–49], median [IQR]). Demographic, clinical and genetic characteristics of both cohorts are detailed in Table 1. All subjects provided written informed consent prior to participation according to the Declaration of Helsinki. The study was approved by the local ethics committees of the University of Tübingen and the other study sites.

**Genotyping of SCA3 subjects**

The repeat length of the expanded and normal alleles was determined by PCR-based fragment length analysis from 100 to 250 ng genomic DNA (CEQ8000 capillary sequencer, Beckman Coulter). CAG repeat length was determined in a centralised manner for cohort #1.

**Human serum and neuropathology**

Blood samples were centrifuged (4,000 $g$, 10 min, room temperature). Serum was frozen at $-80°C$ within 60 min after collection, shipped and analysed without any previous thaw–freeze cycle. Post-mortem brain and spinal cord tissue of the SCA3 subject ID27925 was assessed for degeneration and polyQ aggregates (Appendix Table S1).

**Mouse model, murine plasma and neuropathology**

We used the 304Q SCA3 knock-in mouse model containing a 304 trinucleotide repeat expansion in the murine *ATXN3* homolog, which was generated on the background of C5BL/6N mice (Charles River) by zinc finger technology (Carbery et al, 2010), as described in detail elsewhere (preprint: Haas et al, 2020; Martier et al, 2019). In short, this method resulted in a double-strand break within the murine (CAACAGCAG)$_2$ region and the introduction of a specific expansion of the CAACAGCAG region accomplished by homologous recombination using a donor vector with (CAACAGCAG)$_{48}$ repeats, flanked by 800 bp up- and downstream of *ATXN3* (for a more detailed description of the mouse model, see preprint: Haas et al (2020)). More repeats than contained in the donor vector were introduced, resulting in the expression of 304 glutamines in the ataxin-3 protein under control of all endogenous regulatory elements. The length of the expansion was chosen to trigger a more pronounced behavioural phenotype than observed in other knock-in SCA3 models, where the polyQ expansion length is in the range of SCA3 patients, but which display only mild or no phenotypes (Ramani et al, 2015, 2017; Switonski et al, 2015). We used an interrupted CAG repeat (interruptions of the CAG repeat by CAA, i.e. (CAGCAGCAA)$_n$) as (i) this allowed to ensure meiotic stability of the repeat length across generations (i.e. to keep the repeat length stable) (Frontali et al, 1999), and as (ii) an interrupted repeat is also found in the murine ataxin-3 locus. We genotyped animals ($n = 147$) by PCR, using DNA extracted from ear biopsies with the High Pure PCR Template Preparation Kit (Roche).

**Table 1. Demographic, clinical and genetic characteristics of the human SCA3 cohorts.**

| Cohort #1 (ESMI) | Ataxic SCA3 | Preataxic SCA3 | Controls |
|---|---|---|---|
| Sample size (female) | 75 (56.0%) | 8 (87.5%) | 77 (41.6%) |
| Age (years) | 52.3 (45.7–61.0) | 33.7 (30.1–36.9) | 43.5 (31.4–58.9) |
| Reported age of onset (years) | 40.0 (33.0–46.5) | NA | NA |
| Disease duration based on reported age of onset (years) | 11.0 (7.2–15.1) | NA | NA |
| SARA score | 13.0 (10.0–21.5) | 1.0 (0.5–2.0) | NA |
| Repeat count (long allele) | 70 (67–71) | 70 (67–71) | NA |
| Predicted time to onset (years) | NA | 9.1 (7.7–13.4) | NA |
| **Cohort #2 (EuroSCA/RiSCA)** | **Ataxic SCA3** | **Preataxic SCA3** | **Controls** |
| Sample size (female) | 27 (48.1%) | 14 (42.9%) | 48 (52.1%) |
| Age (years) | 50.9 (46.3–55.0) | 36.7 (28.2–45.5) | 40.7 (31.2–50.0) |
| Reported age of onset (years) | 40.0 (35.3–46.0) | NA | NA |
| Disease duration based on reported age of onset (years) | 10.9 (6.1–16.9) | NA | NA |
| SARA score | 11.5 (6.5–15.0) | 1.0 (0.5–1.5) | NA |
| Repeat count (long allele) | 69 (66–70) | 69 (65–70) | NA |
| Predicted time to onset (years) | NA | 9.8 (8.8–14.3) | NA |

Data are reported as median and interquartile range.
NA, not applicable.

The study comprised 147 animals (56 heterozygous, 37 homozygous and 54 wild-type) with balanced sex within the genotypic groups. For the primary analysis, we aimed to sample $\geq 10$ animals per genotype (heterozygous/wild-type) and age group (2/6/12/18 months), based on effect sizes observed in human studies (Wilke et al, 2018) and in previous murine studies (Bacioglu et al, 2016); no animals were excluded from the analyses. Within each genotype, animals were randomly allocated to an age group (2, 6, 12 or 18 months of age). Animals were dissected following $CO_2$ introduction. Blood was obtained by heart puncture with heparinised syringes, collected in 1.3 ml lithium heparin tubes and centrifuged (3,000 $g$, 20 min, 4°C) to obtain plasma. Samples were then aliquoted (70 µl) and stored at −80°C until use. Cerebellum and frontal lobe tissue were homogenised in Precellys CK14 tubes on the Precellys 24 homogeniser (VWR) at 10% (w/v) in homogenisation buffer (50 mM Tris [pH 8.0], 150 mM NaCl, 5 mM EDTA, containing proteinase inhibitor [cOmplete Protease Inhibitor Cocktail, Roche]). Cerebellar samples were further treated with ultrasonic waves for better homogenisation. The homogenised brain tissue was centrifuged at 25,000 $g$, 4°C, for 60 min. The supernatant was aliquoted and diluted 1:10,000 in sample diluent. We quantified tissue levels of soluble mutant ataxin-3 by time-resolved Förster resonance energy transfer immunoassay (TR-FRET) (Nguyen et al, 2013) and aggregated mutant ataxin-3 by filter retardation assay (Weber et al, 2017; Weishäupl et al, 2019), as detailed below.

## Murine histology and immunohistochemistry

The mice sampled for these analyses were transcardially perfused with cold PBS and 4% paraformaldehyde (PFA). Brains were post-fixed in 4% PFA overnight at 4°C before paraffin embedding. Seven-micrometer sagittal sections were cut with the Leica RM2155 microtome (Leica, Germany). Before staining, sections were rehydrated in xylene and a graded alcohol series with the Leica Autostainer XL. For immunohistochemistry, microwave treatment with 10 mM sodium citrate and 10 mM citric acid for 15 min and threefold washing with phosphate-buffered saline (PBS) were performed. Endogenous peroxidase was blocked by 1.6% peroxidase (Sigma-Aldrich, Germany). After washing with PBS, sections were blocked in 5% normal goat serum (NGS; Vector Laboratories, USA) in PBS supplemented with 0.3% Triton X-100 (Carl Roth, Germany) at room temperature for 45 min. After washing with PBS, sections were incubated with primary antibodies: mouse anti-ataxin-3 (clone 1H9, 1:500, MerckMillipore, Germany), mouse anti-calbindin (D28-K, 1:1,000, Swant, Switzerland), mouse anti-neurofilament L (NfL, clone NFL3, 1:500), and mouse anti-neurofilament H and phosphorylated anti-neurofilament H (clone SMI31, 1:1,000, both from Biolegend, USA), diluted in PBS containing 15% NGS overnight at 4°C in a humid chamber. After washing with PBS, biotinylated secondary antibody (goat anti-mouse, 1:200, Vector Laboratories, Burlingame, USA) was incubated on the sections for 1 h at room temperature. Afterwards, sections were incubated with avidin–biotin complex (ABC; Vector Laboratories, USA) for 1 h at room temperature. After washing, the substrate 3,3′-Diaminobenzidin (DAB, Sigma-Aldrich, Munich, Germany) was added to the sections and the reaction was stopped in distilled water. After dehydrating, the sections were mounted with CV Ultra Mounting Media (Leica, Germany). For cresyl violet (Nissl) staining, rehydrated sections were incubated in 0.1% Cresyl Violet Stain Solution (ab246816, Abcam, UK) for 5 min at room temperature. Afterwards, sections were shortly rinsed in distilled water to reduce background noise and dehydrated in alcohol and xylene before mounting with CV Ultra Mounting Media.

Imaging of the sections was performed using the Axioplan2 imaging microscope with an AxioCam HR colour digital camera and the AxioVision SE64 Rel. 4.9 software (Zeiss, Germany). All quantifications of cell number were performed in 2 mice per genotype and age and 4 different sections (= cerebellar loops) per mouse, with the examiner being blinded to genotype and age. Size of deep cerebellar nuclei was quantified from cresyl violet-stained sections using AxioVision SE64 Rel. 4.9 software. Cell number of pontine nuclei and frontal cortex was assessed on the basis of cresyl violet staining of 3 visual fields at 20× magnification using ImageJ (imagej.nih.gov). Briefly, colour images were converted to 16-bit grey-scale and threshold was adjusted to highlight all structures. To count cells automatically, the function "analyse particles" in ImageJ was used. Purkinje cells (PCs) were quantified and classified manually into the categories "altered" (= shrinkage of PC soma), "lost" (= no visible PC soma) and "intact" based on NfL and Nissl-stained cerebellar sections.

## Phenotypic assessment of mice

Given the strong abnormal weight phenotype of our SCA3 mouse model, indicating severe underlying disease processes already early in the disease course, weight was determined longitudinally in all animals every 2 weeks. Additionally, in an independent cohort of 304Q SCA3 mice ($n = 45$), evolution of coordination and balance was assessed by quantitative gait analysis using the Catwalk 8.1 gait analysis system (Noldus). Animals were tested every 6 months starting with 6 months of age, 12–15 mice per genotype were analysed per time point. Each mouse had to complete five runs with a minimal run duration of 0.5 s and maximum of 10 s. Speed variation had to be below 60%. Run performance and footprint analysis were performed with Catwalk X software (Noldus).

The experiments were conducted in accordance with the veterinary office regulations of Baden–Wuerttemberg, Germany (Regierungspräsidium Tübingen: HG3/13, HG2/18), the German Animal Welfare Act and the guidelines of the Federation of European Laboratory Animal Science Associations, based on European Union legislation (directive 2010/63/EU). Animals were housed under standard conditions in groups ($\leq 5$ per cage) and maintained within a 12-h light–dark cycle. Access to food and water was given ad libitum.

## Neurofilament quantification

Neurofilament concentrations were measured in duplicates by ultra-sensitive single-molecule array (Simoa) technique on the Simoa HD-1 Analyzer (Quanterix, Lexington, Massachusetts). In cohort #1, we used the NF-light Advantage Kit for NfL (Kuhle et al, 2016) and the pNF-heavy Discovery Kit for pNfH quantification (Wilke et al, 2019), according to the manufacturer's instructions (Quanterix). In cohort #2, we quantified Nf concentrations using an independent homebrew NfL-pNfH duplex assay, as detailed below. Given the minute volumes that suffice for measuring both analytes by the NfL-pNfH duplex assay (70 µl per sample), we also used this duplex assay for quantifying Nf concentrations in murine plasma and

murine brain lysates from cerebellum and frontal lobe. All samples were measured in duplicates (dilution for serum/plasma: 1:4, for lysates: 1:2,000). Technicians were blinded to the genotypic and phenotypic status of the samples. Validation with 47 independent serum samples (13 SCA3 subjects, 34 controls) confirmed excellent agreement between the Quanterix and the homebrew NfL assays ($R = 0.99$). This allowed transformation of the NfL homebrew measurements to the scale of the Quanterix measurements by linear regression (see Appendix Table S2 for assay characteristics and validation). For pNfH, analogous validation also showed good, yet lower, agreement between the Quanterix and the homebrew pNfH assays ($R = 0.88$).

### Homebrew NfL-pNfH duplex Simoa assay

Neurofilament light and pNfH levels in cohort #2 were measured by an in-house multiplex Simoa assay. For NfL, the monoclonal antibody (mAB) 47:3 was used as capture antibody and mAB 2:1 as detector antibody (UmanDiagnostics, Umeå, Sweden). pNfH was captured by mouse mAB anti-human pNfH (Iron Horse Diagnostics, USA) and detected by chicken polyclonal anti-human pNfH antibodies (Iron Horse Diagnostics). Each capture antibody was separately coupled to dye-encoded paramagnetic beads using a procedure previously described (Disanto *et al*, 2017). Bovine-lyophilised NfL was obtained from UmanDiagnostics and purified pNfH from Iron Horse. Calibrators ranged for NfL from 0 to 2,000 pg/ml, and for pNfH from 0 to 400 pg/ml (calibrator diluent: tris base saline [TBS]; 0.1% Tween-20; 1% milk powder; 300 µg/ml Heteroblock [Omega Biologicals Inc., Bozeman, USA]). Batch prepared calibrators were stored at −80°C. Calibrators (neat) and samples (serum/plasma 1:4 dilution, lysates 1:2,000 dilution; sample diluent: TBS, 0.1% Tween-20, 1% milk powder, 400 µg/ml Heteroblock [Omega Biologicals]) were measured in duplicates. Reagents were prepared as follows: NfL and pNfH beads were diluted to $1 \times 10^4$ beads/µl each, and detector antibodies for NfL and pNfH were adjusted to 0.1 and 0.05 µg/ml, respectively (bead and detector diluent: TBS; 0.1% Tween-20; 1% milk powder; 300 µg/ml Heteroblock [Omega Biologicals]). Parallelism of the assay for serum, plasma and brain lysates was confirmed by serial dilution experiments in native samples.

### Quantification of soluble ataxin-3

Soluble mutant ataxin-3 measurements via time-resolved Förster resonance energy transfer immunoassay (TR-FRET) were performed using a combination of an anti-ataxin-3 antibody (clone 1H9, MAB5360, Merck) labelled with Tb (donor) fluorophore and an anti-polyQ antibody (clone 1C2, MAB1574, Merck) labelled with d2 (acceptor) fluorophore (labelling by Cisbio). Briefly, homogenised cerebellar and frontal lobe samples for Nf measurements were diluted in homogenisation buffer (50 mM Tris, 150 mM NaCl, 5 mM EDTA, cOmplete Protease Inhibitor Cocktail) to a final concentration of 1 µg/µl total protein amount. Next, 5 µl of diluted sample was incubated with 1 µl of the TR-FRET antibody mix (1 ng 1H9-Tb + 3 ng MW1-d2 in 50 mM NaH2PO4, 400 mM NaF, 0.1% BSA, 0.05% Tween-20) in a low-volume white ProxiPlate 384 TC Plus plate (PerkinElmer) at 4°C for 22 h. Signals were detected at 620 nm and 665 nm using an EnVision Multimode Plate Reader with a TRF-laser unit (PerkinElmer).

### Quantification of aggregated ataxin-3

For the filter retardation assay, 12.5 µg of protein homogenate was diluted in Dulbecco's phosphate-buffered saline (DPBS; Gibco) containing 2% sodium dodecyl sulphate (SDS) and 50 mM 1,4-dithiothreitol (DTT) and heat-denatured for 5 min at 95°C. Samples were filtered through Amersham Protran Premium 0.45-µm nitrocellulose membranes (GE Healthcare) by using a Minifold II Slot-Blot System (Schleicher & Schuell). Before loading the samples, the membrane was equilibrated with DPBS containing 0.1% SDS and rinsed afterwards twice with DPBS. The membranes were blocked in tris-buffered saline (TBS; 1 M Tris, 5 M NaCl) containing 5% skimmed milk powder (Sigma-Aldrich) for 1 h, followed by primary antibody (mouse anti-ataxin-3, 1:2,500, clone 1H9, MAB5360, Merck Millipore) incubation overnight at 4°C, and secondary antibody (IRDye 800CW goat anti-mouse IgG (H+L) 1:1,000, 926-32210, LI-COR) incubation for 1.5 h at room temperature. Detection and quantification of the fluorescent signal was performed using the ODYSSEY Server software version 4.1 (LI-COR Bioscience).

### Statistical analysis

#### *Group effects*
We used non-parametric procedures to analyse group effects on Nf levels (Mann–Whitney *U*-tests, two-sided, Bonferroni-corrected for multiple comparisons). To correct the group effects for age-dependent increases of Nf levels (log-transformed) within each cohort, we used linear models with the factors group, age and their interaction. The optimal cut-offs for differentiating ataxic SCA3 subjects from controls by their Nf levels were determined according to Youden's procedure, with the cut-off allowing to benchmark Nf levels in preataxic SCA3 subjects.

#### *Associations between disease severity and disease progression*
In ataxic subjects, we assessed the association between Nf levels (log-transformed) and disease severity, as captured by the SARA score, with Pearson's correlations (two-sided test). We used partial correlations with the covariates age, disease duration and CAG count to adjust the association between Nfs and disease severity for potential confounders. Prospective longitudinal SARA scores were available for a subset of patients ($n = 35$; all from cohort #2; number of longitudinal visits per subject: 2–4) allowing us to determine intraindividual disease progression as the annual change of SARA scores (determined by intraindividual linear regression coefficients, using all available SARA scores).

#### *Association of NfL levels with age and repeat length*
We analysed the association of NfL levels with age and CAG repeat length in SCA3 carriers with a linear model, using the pooled NfL data of both cohorts after transforming NfL levels of cohort #2 (which were measured by homebrew Simoa) to the scale of the Quanterix Simoa used for cohort #1. Specifically, we modelled NfL levels (log-transformed) in all SCA3 carriers ($n = 123$) with the predictors age and *ATXN3* CAG repeat length, their squares and all possible interactions, analogous to previous analyses in Huntington's disease (Byrne *et al*, 2017). We centred age at 50 years (i.e. mean age of carriers) and CAG repeat length at 68 (i.e. mean CAG repeat length of carriers), as in analogous previous analyses (Byrne

*et al*, 2017). We excluded one outlier (NfL > 700 pg/ml) to fulfil model assumptions.

### Association of NfL with time to onset in preataxic SCA3

We analysed NfL levels as a function of time to estimated onset of ataxia in preataxic SCA3 by linear regression. For each preataxic SCA3 subject, we individually calculated the time to the estimated onset of ataxia based on CAG repeat count and the age at the time of assessment, as established previously (Tezenas du Montcel *et al*, 2014). Hereby, the estimate based on the repeat size is adjusted for the age which the individual has actually reached without developing ataxia; i.e., the older the preataxic subject at the time of assessment, the higher the predicted age at onset. To determine the point of time at which NfL levels become significantly increased in the preataxic stage of SCA3, NfL levels of preataxic carriers needed to be related to the NfL levels of controls at the same age, as NfL levels physiologically increase with age in controls (Wilke *et al*, 2016). Hence, we calculated the *z*-score of each SCA3 subject in relation to the NfL distribution in controls at the same age (for a visualisation, see Fig 3B). For this, the difference between the measured NfL level and the NfL level estimated for controls at the same age was standardised relative to the NfL distribution in controls at this age. NfL in controls was modelled by linear regression on the level of log-transformed data.

### Intraindividual stability of Nf levels

We assessed intraindividual stability of Nf levels by calculating the intraclass correlation coefficient (ICC) of each analyte (model specification: two-way mixed, single-measures, absolute agreement), using our 6-week longitudinal Nf data (log-transformed) from a subset of subjects ($n = 21$) of cohort #2 where such 6-week interval sample was available.

### Sample size estimation for future treatment trials

We performed sample size estimations for future intervention trials, which use the reduction of Nf levels towards levels observed in healthy controls as outcome measure (Byrne *et al*, 2018). We estimated the sample size per study arm, which would be required to detect a given control-adjusted relative reduction of Nf levels (10–70%) in the treatment arm, assuming that null mean change over time occurred in the placebo arm of the trial. To this end, we based the assumed intersubject variability in the hypothetical trial on the measured intrasubject variability in the change of analyte levels (from baseline to 6 weeks) in our SCA3 subjects (Fig 4). The estimation further assumed equal numbers of subjects in both study arms, $\alpha = 0.01$, $\beta = 0.01$, two-tailed independent *t*-tests, and the use of log-transformed analyte levels. It was performed with GPower 3.1 software (Kiel, Germany).

### Murine data

We compared biomarker levels (NfL, pNfH, aggregated ataxin-3, soluble ataxin-3) and phenotypic data between heterozygous and wild-type animals within the same age group (i.e. animals sacrificed at 2, 6, 12 or > 18 months), using independent *t*-tests (two-sided, Bonferroni-corrected for multiple comparisons given that two analytes were measured from each sample). Analogously, data were compared between homozygous mutants and wild-type animals.

Throughout the manuscript, if the assumption of normality was violated (assessed by inspection and Shapiro–Wilk test), we used log-transformed data for the statistical analysis after ensuring that the transformed data were normally distributed. If normality was violated also after transformation, non-parametric tests were applied. For parametric tests, corrections were applied if variance differed between the groups that were compared (*t*-tests adjusted for unequal variances). We analysed the data with SPSS (IBM, version 24). We reported the effect size *r* of the statistical tests wherever possible.

## Data availability

All source data of this study are available in the supplementary material of the article.

**Expanded View** for this article is available online.

### Acknowledgements

Biosamples were obtained from the Neuro-Biobank of the University of Tübingen, Germany, which is supported by the University of Tübingen, the Hertie Institute for Clinical Brain Research (HIH) and the German Center for Neurodegenerative Diseases (DZNE). CW, HH, BW, AD, TK, RS, LS and MS are members of the European Reference Network for Rare Neurological Diseases Project ID No 739510. This work was supported by the Horizon 2020 research and innovation programme (grant 779257 Solve-RD to MS and RS), the National Ataxia Foundation (grant to CW and MS), the Wilhelm Vaillant Stiftung (grant to CW), the EU Joint Programme—Neurodegenerative Disease Research (JPND) through participating national funding agencies, and the European Union's Horizon 2020 research and innovation programme under grant agreement No 643417. BM was supported in part from the grant NKFIH 119540. HJ was funded by the Medical Faculty of the University of Heidelberg. CB was funded by the University of Basel (PhD Program in Health Sciences). The funding sources had no role in the study design, data collection, data analysis, data interpretation or writing of the manuscript.

### Author contributions

CW design and conceptualisation of the study, acquisition of data, analysis of the data, and drafting and revision of the manuscript. EH design and conceptualisation of the study, conduction of animal experiments and acquisition of data, analysis of the data and revision of the manuscript. KR design and conceptualisation of the study, subject recruitment, acquisition of data and revision of the manuscript. JF, HG-M, MMS, BW, HH, ML, AF, AD, BM, MM, JI, PG, JV, LPA, MR, HJ and LS subject recruitment, acquisition of data and revision of the manuscript. MN acquisition of neuropathological data, analysis of the data and revision of the manuscript. RS and JK acquisition of data and revision of the manuscript. SAK design and conceptualisation of the study and revision of the manuscript. TK design and conceptualisation of the study, subject recruitment, acquisition of data and revision of the manuscript. CB design and conceptualisation of the study, acquisition of data, analysis of the data and revision of the manuscript. JH-S design and conceptualisation of the study, conduction of animal experiments and acquisition of data, analysis of the data and revision of the manuscript. MS design and conceptualisation of the study, acquisition of data, analysis of the data, and drafting and revision of the manuscript.

## The paper explained

**Problem**

Spinocerebellar ataxia type 3 (SCA3), the most frequent autosomal-dominant ataxia worldwide, is a prototypic neurodegenerative repeat-expansion disorder. As targeted molecular treatments for SCA3 (e.g. antisense oligonucleotides) are coming into reach, easily accessible peripheral biomarkers are warranted, both for human and for preclinical trials. Such biomarkers are particularly important at the presymptomatic disease stage, where disease-modifying therapies might be most effective, and require cross-validation in animal models, as well as cross-validation with associated central nervous system changes.

**Results**

In two independent multicentric human SCA3 cohorts, blood levels of neurofilament light (NfL) and phosphorylated neurofilament heavy (pNfH) were each increased at the symptomatic disease stage. NfL levels were increased also at the presymptomatic stage. NfL elevations were present already 7.5 years before the individual expected symptom onset, with levels increasing further in proximity to the conversion from the presymptomatic to the symptomatic disease stage. NfL levels reflected both subjects' clinical disease severity and disease progression. The neurofilament increases in our human cohorts were paralleled by neurofilament increases in a SCA3 knock-in mouse model, here also starting already at the presymptomatic stage, closely following the onset of ataxin-3 aggregation and preceding significant Purkinje cell loss in the brain. These results allowed mapping a larger biomarker cascade of SCA3 disease, capturing differential changes in NfL, pNfH, ataxin-3 and behavioural biomarker trajectories across disease stages.

**Impact**

Blood levels of neurofilaments, particularly NfL, might provide easily accessible peripheral biomarkers of neuronal damage in SCA3, validated both at the presymptomatic and at symptomatic disease stage in humans and mice and associated with brain pathology changes already at the earliest disease stages. NfL levels might serve as progression, proximity-to-onset and, potentially, treatment-response biomarkers for both human and preclinical trials.

## Conflict of interest

BW receives research funding from Radboud university medical centre, ZonMW, Hersenstichting and Gossweiler Foundation. MR receives funding from the Polish National Center for Research and Development (grant IS-2/230/NCBR/2015). TK receives/has received research support from the Deutsche Forschungsgemeinschaft (DFG), the German Federal Ministry of Education and Research (BMBF), the German Federal Ministry of Health (BMG), the Robert Bosch Foundation, the European Union (EU), and the National Institutes of Health (NIH). He has received consulting fees from Biohaven and UBC. He has received a speaker honorarium from Novartis. MS received speaker's honoraria and research support from Actelion Pharmaceuticals, unrelated to the current project and manuscript. The other authors declare no competing financial interests.

## For more information

(i)     Information on the National Ataxia Foundation (NAF): https://ataxia.org/what-is-ataxia/

(ii)    Genetic information on spinocerebellar ataxia type 3, provided by Online Mendelian Inheritance in Man (OMIM): https://omim.org/entry/109150

(iii)   Clinical information on spinocerebellar ataxia type 3, provided by US National Library of Medicine, Genetics Home Reference: https://ghr.nlm.nih.gov/condition/spinocerebellar-ataxia-type-3

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

**Appendix 1**

For the SCA3 neurofilament study group: Christian Deuschle[1,2], Elke Stransky[1], Kathrin Brockmann[1,2], Jörg B Schulz[3,4], Laszlo Baliko[5], Judith van Gaalen[6], Mafalda Raposo[7], Andreas Jeromin[8]

[1]Hertie Institute for Clinical Brain Research (HIH) and Center of Neurology, University of Tübingen, Tübingen, Germany

[2]German Center for Neurodegenerative Diseases (DZNE), University of Tübingen, Tübingen, Germany

[3]Department of Neurology, RWTH Aachen University, Aachen, Germany

[4]JARA-BRAIN Institute Molecular Neuroscience and Neuroimaging, Forschungszentrum Jülich, RWTH Aachen University, Aachen, Germany

[5]Department of Medical Genetics, and Szentagothai Research Center, University of Pécs Medical School, Pécs, Hungary

[6]Donders Institute for Brain, Cognition, and Behaviour, Department of Neurology, Radboud University Medical Center, Nijmegen, The Netherlands

[7]Faculdade de Ciências e Tecnologia, Universidade dos Açores, Ponta Delgada, Portugal

[8]Quanterix Corporation, Lexington, KY, USA

