## [Review Process File · EMBO Molecular Medicine]

Neurofilaments in spinocerebellar ataxia type 3: blood biomarkers at the preataxic and ataxic stage in humans and mice

Carlo Wilke, Eva Haas, Kathrin Reetz, Jennifer Faber, Hector Garcia-Moreno, Magda Santana, Bart van de Warrenburg, Holger Hengel, Manuela Lima, Alessandro Filla, Alexandra Durr, Bela Melegh, Marcella Masciullo, John Infante, Paola Giunti, Manuela Neumann, Jeroen de Vries, Luís Pereira de Almeida, Maria Rakowicz, Heike Jacobi, Rebecca Schüle, Stephan Käser, Jens Kuhle, Thomas Klockgether, Ludger Schöls, Christian Barro, Jeannette Hübener-Schmid, and Matthias Synofzik

DOI: 10.15252/emmm.201911803

Corresponding author(s): Matthias Synofzik (matthis.synofzik@uni-tuebingen.de)

Review Timeline:

Submission Date:	22nd Nov 19
Editorial Decision:	20th Dec 19
Revision Received:	1st Apr 20
Editorial Decision:	23rd Apr 20
Revision Received:	5th May 20
Accepted:	10th May 20

Editor: Celine Carret

Transaction Report:

20th Dec 2019

Dear Prof. Synofzik,

Thank you for the submission of your manuscript to EMBO Molecular Medicine. We have now heard back from the three referees whom we asked to evaluate your manuscript.

While overall the referees find the study to be of interest and important for the field, you will see that they also found issues that must be addressed in a revised version of your article. An important issue is novelty in neurodegenerative diseases but also in SCA3. While the provision of the mouse model increases the novelty of the study, we feel that providing more details about its generation and characterisation within this article (ref. 2) is required. Also, using this in vivo setting to provide some mechanism (ref. 3) would be a very nice and welcome addition, which would also increase the novelty of the findings. Further, more details and explanations and a better discussion are needed, as well as new analyses warranted (like females vs. males - ref. 3).

We would therefore welcome the submission of a revised version within three months for further consideration and would like to encourage you to address all the criticisms raised as suggested to improve conclusiveness and clarity. Please note that EMBO Molecular Medicine strongly supports a single round of revision and that, as acceptance or rejection of the manuscript will depend on another round of review, your responses should be as complete as possible.

I look forward to receiving your revised manuscript.

Happy Holidays!

Celine Carret

Celine Carret, PhD
Senior Editor
EMBO Molecular Medicine

*** Instructions to submit your revised manuscript ***

** PLEASE NOTE ** As part of the EMBO Publications transparent editorial process initiative (see our Editorial at <https://www.embopress.org/doi/pdf/10.1002/emmm.201000094>), EMBO Molecular Medicine will publish online a Review Process File to accompany accepted manuscripts.

To submit your manuscript, please follow this link:

(Link Not Available)

- 1) a .doc formatted version of the manuscript text (including Figure legends and tables). Please make sure that the changes are highlighted to be clearly visible to referees and editors alike.
- 2) separate figure files*
- 3) supplemental information as Expanded View and/or Appendix. Please carefully check the authors guidelines for formatting Expanded view and Appendix figures and tables at <https://www.embopress.org/page/journal/17574684/authorguide#expandedview>
- 4) a letter INCLUDING the reviewers' reports and your detailed responses to their comments (as Word file)

Also, and to save some time should your paper be accepted, please read below for additional information regarding some features of our research articles:

- 5) The paper explained: EMBO Molecular Medicine articles are accompanied by a summary of the articles to emphasize the major findings in the paper and their medical implications for the non-specialist reader. Please provide a draft summary of your article highlighting
 - the medical issue you are addressing,
 - the results obtained and
 - their clinical impact.

- 6) For more information: There is space at the end of each article to list relevant web links for

further consultation by our readers. Could you identify some relevant ones and provide such information as well? Some examples are patient associations, relevant databases, OMIM/proteins/genes links, author's websites, etc...

7) Author contributions: the contribution of every author must be detailed in a separate section (before the acknowledgments).

8) EMBO Molecular Medicine now requires a complete author checklist (<https://www.embopress.org/page/journal/17574684/authorguide>) to be submitted with all revised manuscripts. Please use the checklist as a guideline for the sort of information we need WITHIN the manuscript as well as in the checklist. This is particularly important for animal reporting, antibody dilutions (missing) and exact p-values and n that should be indicated instead of a range.

9) Every published paper now includes a 'Synopsis' to further enhance discoverability. Synopses are displayed on the journal webpage and are freely accessible to all readers. They include a short stand first (maximum of 300 characters, including space) as well as 2-5 one sentence bullet points that summarise the paper. Please write the bullet points to summarise the key NEW findings. They should be designed to be complementary to the abstract - i.e. not repeat the same text. We encourage inclusion of key acronyms and quantitative information (maximum of 30 words / bullet point). Please use the passive voice. Please attach these in a separate file or send them by email, we will incorporate them accordingly.

You are also welcome to suggest a striking image or visual abstract to illustrate your article. If you do please provide a jpeg file 550 px-wide x 400-px high.

10) A Conflict of Interest statement should be provided in the main text

11) Please note that we now mandate that all corresponding authors list an ORCID digital identifier. This takes <90 seconds to complete. We encourage all authors to supply an ORCID identifier, which will be linked to their name for unambiguous name identification.

Currently, our records indicate that there is no ORCID associated with your account.

Please click the link below to provide an ORCID:
(Link Not Available)

12) The system will prompt you to fill in your funding and payment information. This will allow Wiley to send you a quote for the article processing charge (APC) in case of acceptance. This quote takes into account any reduction or fee waivers that you may be eligible for. Authors do not need to pay any fees before their manuscript is accepted and transferred to our publisher.

Each figure should be given in a separate file and should have the following resolution:
Graphs 800-1,200 DPI
Photos 400-800 DPI
Colour (only CMYK) 300-400 DPI"

*Additional important information regarding figures and illustrations can be found at <http://bit.ly/EMBOPressFigurePreparationGuideline>

***** Reviewer's comments *****

Referee #1 (Comments on Novelty/Model System for Author):

The novelty aspect is lowered by this recent publication, which includes many of the key findings:

Mol Neurodegener. 2019 Nov 4;14(1):39. doi: 10.1186/s13024-019-0338-0.

Neurofilament light chain is a promising serum biomarker in spinocerebellar ataxia type 3.

Li Q et al

Referee #1 (Remarks for Author):

This is a paper that nicely demonstrates that patients with SCA3 have increased levels of neurofilament light and (to a lesser degree) neurofilament heavy chain in serum. The NfL levels correlated to symptoms and CAG-repeats, but were increased already in "preataxic" stage in many patients. The biomarker findings were robust and reproduced in two different cohorts, with two different assays for NFL. For NFL, the findings seem to partly reproduce the recent findings by Li et al in Mol Neurodegener 2019; 14(1):39. The authors also translate their findings to an animal model of the disease. Correlations were also described with neuropathological findings in one individual.

The evidence appears quite solid that serum/plasma levels of NfL is a promising biomarker to track neurodegeneration in SCA3. The translational consistency between human patients (including in preclinical stage) and an animal model is encouraging. The findings may be useful both to study the disease mechanisms, monitor patients in clinical practice, and to facilitate drug development.

I only have minor comments to the paper:

Longitudinal SARA scores:

The authors found that serum NfL correlated with change in SARA scores. Was this correlation also found when adjusting for baseline SARA score?

Figure 4: The term "Temporal dynamics" is used in a description of the relationship between cross-sectional NfL and time from expected onset. The term "temporal dynamics" may be slightly misleading here, since it may lead to the belief that longitudinal NfL levels (or longitudinal symptoms) were measured. I suggest to change the terminology to avoid these misunderstandings.

Referee #2 (Remarks for Author):

The manuscript provides convincing experimental evidence demonstrating that the increase in the

serum levels of neurofilament light chain (NfL) can be used as biomarker at the preataxic and ataxic stage of spinocerebellar ataxia type 3. The increase in NfL has been previously associated with axonal damage and is a promising biomarker for both academic research and clinical diagnosis/prognosis for several neurological disorders. Preliminary studies by the authors with a smaller cohort had already suggested an increase in serum NfL levels of SCA3 patients, but the current study significantly expands those findings and also includes data on the variation in serum levels of pNfH. An important strength of this manuscript is the demonstration that the increase in serum NfL levels observed in pre-ataxic and ataxic SCA3 is mirrored in the SCA3 knock-in mice model. This is an important finding that will be extremely important for a large scientific community in urgent need for relevant biomarkers to follow disease progression as well as response to therapeutic SCA3 trials in both clinical and preclinical settings. The manuscript is well written with a clear explanation of the relevant experimental details, statistical analysis and models developed.

Relevant information stands out from this manuscript that is of interest for the broad scientific and clinical community interested in neurological diseases in general and SCA3. In particular:

- 1) the evidence that serum NfL levels increase with repeat length and patient's age;
- 2) data showing that NfL levels increase with the proximity of the expected disease onset and significant increase occurs ~7.5 years before onset;
- 3) the temporal analysis of NfL and pNfH in the mouse model showing that these biomarkers reveal changes prior to weight and motor phenotypes.

Major issues:

1) The details about the generation and full characterization of the mouse knock-in model used in for the cross-species biomarker evaluation is not presented in the manuscript and reported as "described in detail elsewhere (Martier et al.) (Haas et al., in preparation)". The reference Martier et al. describes the use of the knock-in mouse model but does not show details on the production and characterization of this SCA3 disease model. It would be important for the scientific community to have complete information/characterization about the knock-in mouse model used for biomarker analysis and this data should be made available before the publication of the current manuscript.

Minor points:

1) It is evident that this study presents a thorough experimental analysis and expands on previous studies by the authors and others (e.g. Wilke et al, 2018; Li et al, 2019). However, it should be better clarified throughout the manuscript the common findings and eventual differences in biochemical parameters (NfL levels encountered) and neuropathologic observations in comparison with the study of Li et al. (2019) that includes two cohorts with 26 and 207 individuals, respectively.

2) Please include reference to previous study by Li et al. 2019 in the introduction (second paragraph, Page 4)

3) Are the the neuropathologic finding for SCA3 subject ID27925 somehow comparable with those of Li et al. 2019 (page 10)?

4) Please include more information that compares this study with that of Li et. al, 2019 to the discussion (page 14).

5) In the introduction on page 4, the authors write "We therefore assessed plasma NfL and pNfH also in a SCA3 knock-in mouse model (Martier et al.) (Haas et al., in preparation) across presymptomatic and symptomatic disease stages)". Please complete reference Martier et al. (year

and issue/page numbers) in the manuscript text and in the reference list. If the Haas et al. manuscript is still in preparation, it should be removed from the text and the correct and/or complete reference for the characterization of SCA3 knock-in mice added in replacement.

Referee #3 (Comments on Novelty/Model System for Author):

The paper is well written and NfL is potentially a useful blood biomarker, although it has been used as a biomarker in other neurodegenerative disorders. Thus, the scientific impact is not high, especially regarding innovativeness. The scientific premise of the study is weak for SCA3 despite the knock-in mouse data. I am not sure whether the mouse model is a good model of human SCA3, and what they investigated was essentially histopathological correlation of their NfL data, verifying known histopathological studies of autopsy brains from human SCA3 patients. The mouse study should have added mechanistic understanding to the NfL elevation in blood, but none of these mechanisms were critically investigated. As for a clinical trial readiness, this study provides important information. Nevertheless, the paper will attract a lot of attention from pharmaceutical companies that are interested in this field for therapeutic development.

Referee #3 (Remarks for Author):

The value of this paper is that it provides encouraging data of a badly needed blood biomarker for clinical studies and drug trials for patients with SCA3 and pre-ataxia SCA3-mutation carriers. The use of two independent cohorts with consistent outcomes increases the confidence, and data from the animal model of SCA3 may assure the use of this serum biomarker in preclinical animal studies for the development of disease-modifying drugs for SCA3.

There are some weaknesses and concerns:

1. Blood NfL levels have attracted attentions of researchers as easily accessible biomarkers of various neurodegenerative disorders. Thus, the concept of using blood NfL levels as a biomarker in SCA3 is not particularly novel. Some authors of this paper have published a biomarker paper of NfL in MSA-C, SAOA and SCA1/2/3/6 earlier, and a separate group of investigators reported data on serum NfL levels in SCA3 subjects similar to this paper in November 2019 (Li QF, et al. Neurofilament light chain is a promising serum biomarker in spinocerebellar ataxia type 3. *Mol Neurodegener.* 2019 Nov 4;14(1):39).
2. Neither NfL nor NfH is specific for SCA3. Not only other SCAs but also many other neurological diseases have shown elevated levels. Discussions need to include how the limited specificity should be addressed in clinical trial settings.
3. Discussions should also assess how this will be viewed in the Biomarker Qualification Program of FDA and equivalent standards of the EMA.
4. What biological events do NfL and NfH biomarkers represent? Are neuronal deaths or near deaths required to release neurofilaments? Is cell loss needed? OR is leaky plasma membrane or axonal degeneration sufficient to increase levels of neurofilaments? Authors determine the tissue concentration in the extract of homogenized cerebellum and frontal lobe tissues. However, is there a good correlation between blood and CSF and between CSF and the extracellular compartment in the neurofilament concentration? These scientific premises are important for the interpretation of the presented data. The introduction section should include what are known and unknown about the mechanism/process of NfL and NfH elevation in the blood.
5. It is disappointing that no data regarding #4 are collected in the mouse experiments. Immunofluorescent microscopy/CSF SIMOA to detect NfL/NfH may yield interesting insights. It is particularly valuable if the mouse model shows histological, immunofluorescent, and electron

microscopic changes in the pre-ataxic stage.

6. The lack of longitudinal data does not allow for calculation of the sensitivity to change (SRM).

7. How do NfL and NfH levels compare with the MR biomarkers in terms of sensitivity and specificity?

8. What is the difference in utility between wt/mutant ATXN3 levels and NfL/NfH levels in clinical trials? Can NfL/NfH levels be useful as biomarkers of "target engagement" as authors imply?

9. The repeat that was knocked in contained 304 trinucleotides in the (CAA CAG CAG)₄₈ configuration. The length of 304 trinucleotides is artificially long and may trigger different types of pathogenic mechanisms, such as those mediated by toxic gain of function by (CAG)_{expansion} RNA and repeat-associated non-ATG (RAN) translation. Additionally, the presence of periodic CAA may alter the stability of the expanded repeat in the genomic DNA and binding to RNA-binding proteins by the RNA transcript. While such interrupted expanded repeats have been successfully used in other diseases, investigators of such models took painstaking verifications of the pathogenic authenticity.

10. Motor phenotypes of the SCA3 mouse model should analyze males and females separately since the performance of mice may differ depending on the sex.

11. How does the prediction of the ataxia onset in preataxic mutation carriers compare the prediction based on the CAG repeat size and the current age in the two Cohorts? (See Tezenas du Montcel et al. *J Med Genet*. 2014 Jul;51(7):479-86.)

Reply to comments by the Referees

Referee #1

Remarks for Author: This is a paper that nicely demonstrates that patients with SCA3 have increased levels of neurofilament light and (to a lesser degree) neurofilament heavy chain in serum. The NfL levels correlated to symptoms and CAG-repeats, but were increased already in "preataxic" stage in many patients. The biomarker findings were robust and reproduced in two different cohorts, with two different assays for NFL. For NFL, the findings seem to partly reproduce the recent findings by Li et al in Mol Neurodegener 2019; 14(1):39. The authors also translate their findings to an animal model of the disease. Correlations were also described with neuropathological findings in one individual.

The evidence appears quite solid that serum/plasma levels of NfL is a promising biomarker to track neurodegeneration in SCA3. The translational consistency between human patients (including in preclinical stage) and an animal model is encouraging. The findings may be useful both to study the disease mechanisms, monitor patients in clinical practice, and to facilitate drug development.

Reply: We thank the referee for the positive feedback (regarding the major novelties and differences of our study compared to the study by Li et al. 2019, please see our reply to reviewer #2 point 2).

I only have minor comments to the paper:

1. Longitudinal SARA scores: The authors found that serum NfL correlated with change in SARA scores. Was this correlation also found when adjusting for baseline SARA score?

Reply: We indeed found a significant positive association not only between serum NfL and disease severity ($r=0.43$, $p<.001$, 2-sided test), but also between serum NfL and disease progression ($r=0.34$, $p=0.045$, 2-sided test). This positive association with disease progression was confirmed with a similar effect size ($r=0.29$) when adjusted for baseline SARA score, also showing a statistical trend in this relatively small sub-cohort for which longitudinal SARA scores were available ($p=0.100$, 2-sided test, $n=35$). If significance is confirmed in larger cohorts, this finding suggests that serum NfL levels might be used to stratify SCA3 subjects regarding longitudinal disease progression – in addition to the observed correlation between serum NfL and disease severity.

We now added this SARA-baseline corrected analysis to the Results section.

2. Figure 4: The term "Temporal dynamics" is used in a description of the relationship between cross-sectional NfL and time from expected onset. The term "temporal dynamics" may be slightly misleading here, since it may lead to the belief that longitudinal NfL levels (or longitudinal symptoms) were measured. I suggest to change the terminology to avoid these misunderstandings.

Reply: *The reviewer is right. We rephrased the figure caption in a more descriptive manner as follows (Figure 4): "Association between serum NfL and time from expected onset in preataxic SCA3" (and also rephrased the same term in the Methods section, where it had also previously occurred once).*

Referee #2

The manuscript provides convincing experimental evidence demonstrating that the increase in the serum levels of neurofilament light chain (NfL) can be used as biomarker at the preataxic and ataxic stage of spinocerebellar ataxia type 3. The increase in NfL has been previously associated with axonal damage and is a promising biomarker for both academic research and clinical diagnosis/prognosis for several neurological disorders. Preliminary studies by the authors with a smaller cohort had already suggested an increase in serum NfL levels of SCA3 patients, but the current study significantly expands those findings and also includes data on the variation in serum levels of pNfH. An important strength of this manuscript is the demonstration that the increase in serum NfL levels observed in pre-ataxic and ataxic SCA3 is mirrored in the SCA3 knock-in mice model. This is an important finding that will be extremely important for a large scientific community in urgent need for relevant biomarkers to follow disease progression as well as response to therapeutic SCA3 trials in both clinical and preclinical settings. The manuscript is well written with a clear explanation of the relevant experimental details, statistical analysis and models developed.

Relevant information stands out from this manuscript that is of interest for the broad scientific and clinical community interested in neurological diseases in general and SCA3. In particular:

- 1) the evidence that serum NfL levels increase with repeat length and patient's age;
- 2) data showing that NfL levels increase with the proximity of the expected disease onset and significant increase occurs ~7.5 years before onset;
- 3) the temporal analysis of NfL and pNfH in the mouse model showing that these biomarkers reveal changes prior to weight and motor phenotypes.

Major issues

1. The details about the generation and full characterization of the mouse knock-in model used in for the cross-species biomarker evaluation is not presented in the manuscript and reported as "described in detail elsewhere (Martier et al.) (Haas et al., in preparation)". The reference Martier et al. describes the use of the knock-in mouse model but does not show details on the production and characterization of this SCA3 disease model. It would be important for the scientific community to have complete information/characterization about the knock-in mouse model used for biomarker analysis and this data should be made available before the publication of the current manuscript.

Reply: *We completely agree with the reviewer. We now provide detailed information on the generation and characterisation of the SCA3 knock-in mouse model used for our translational biomarker study by including an extra supplement in the revised version of our manuscript (Appendix Supplementary Methods). In addition, the article which describes the generation and characterisation of the mouse model has now been made available online on bioRxiv (Haas et al. 2020, A novel Ataxin-3 knock-in mouse model mimics the human SCA3 disease phenotype including neuropathological, behavioural, and transcriptional abnormalities. doi:10.1101/2020.02.28.968024). Going beyond this, we now also have added comprehensive neuropathological assessments of the mouse model to the current manuscript (reviewer #3 point 5) and provide additional, sex-specific analyses of the phenotype (reviewer #3 point 10).*

Minor points

2. It is evident that this study presents a thorough experimental analysis and expands on previous studies by the authors and others (e.g. Wilke et al, 2018; Li et al, 2019). However, it should be better clarified throughout the manuscript the common findings and eventual differences in biochemical parameters (NfL levels encountered) and neuropathologic observations in comparison with the study of Li et al. (2019) that includes two cohorts with 26 and 207 individuals, respectively.

Reply: *We appreciate the opportunity to present the commonalities and differences of our present study compared to Wilke et al. 2018 and Li et al. 2019 in more detail. We had already summarised this comparison in our initial manuscript with one dense long sentence, but now realise that this comparison might have been too short and too dense.*

Comparison of the present study with Wilke et al. 2018 and Li et al. 2019

The comprehensive findings of our present study indeed nicely confirm recent preliminary findings with respect to increased NfL blood levels in mono-centric cohorts of ataxic (Wilke et al. 2018, Li et al. 2019) and preataxic (Li et al. 2019) SCA3 subjects, with similar effect sizes. The association of NfL blood levels with clinical disease severity was determined in one sub-cohort

of the Li et al. 2019 study (“cohort B”), indicating a positive association with cross-sectional SARA scores. Our study confirms this association with similar effect size. However, compared to the present study, both of these previous studies show important methodological differences: Both previous studies were only single-centre, single-assay, human-only NfL studies without neuropathology, which were, in addition, limited to cross-sectional data. Moreover, both previous studies lacked the comparative assessment of two biomarkers (i.e. NfL versus pNfH) as well as the biomarker cascade mapping presented here.

Apart from these methodological differences – and even more importantly, compared to both previous studies, our current findings do not only confirm the NfL increase in SCA3 and the association of NfL levels with cross-sectional disease severity, but rather largely extend the previous findings in multiple aspects:

- (1) Our study provides first evidence for a positive association of NfL levels with **longitudinal disease progression** in SCA3, by including first longitudinal SARA scores which allowed assessing individual clinical progression (instead of only disease severity at the moment of blood sampling).
- (2) Our study now provides **first short-term longitudinal NfL and pNfH measurements** in SCA3, which allow assessing intra-individual analyte stability. We acquired short-term longitudinal data in both ataxic and preataxic SCA3 subjects and controls with a sampling interval of 6 weeks (Figure 6). We demonstrate that intra-individual analyte stability in SCA3 is very high for both NfL and pNfH blood levels. Based on this novel finding, we are now able to provide **first sample size estimations for future intervention trials in SCA3** using the reduction of Nf levels as outcome measure. Our data suggest that Nf levels as outcome measures allow a considerable reduction in the sample size required for SCA3 treatment trials compared to clinical scores.
- (3) Our study provides first data in preataxic SCA3 subjects on the **association of the NfL increase with the individual time to expected onset**. This analysis is key to understanding the temporal dynamics of NfL levels in the preataxic stage and to stratifying preataxic subjects with regard to their subclinical neuronal decay.
- (4) Our study moreover demonstrates for the first time that serum levels of **NfL can be used to predict the time to the expected onset of ataxia**, which allows defining a preconversion stage close to the onset of the ataxic stage and helps defining a window of opportunity in which future disease modifying therapies in SCA3 might be most effective.
- (5) Our study assesses **pNfH (phosphorylated neurofilament heavy)** as a novel blood-based biomarker in SCA3, both in the ataxic and preataxic disease stage and both in humans and mice, also allowing us to comparatively assess NfL and pNfH as biomarkers in SCA3. Given the higher sensitivity of NfL to capture changes already at the preataxic stage of human SCA3, our findings provide first evidence to prioritise the use of NfL over pNfH in future SCA3 treatment trials for capturing the preataxic stage. However, both Nfs might

have differential value in capturing different treatment effects and system responses, thus it might even be of value to include **both** of the Nfs investigated here in future treatment trials.

- (6) Our study provides **first associations of Nf levels with neuropathological examinations**, including (a) human neuropathology for hypothesis-generation regarding which neuronal systems may drive the Nf increase in SCA3, and (b) including mouse neuropathology already at the presymptomatic stage, demonstrating that peripheral blood Nf changes are associated with the earliest SCA3 brain processes, closely following ataxin-3 aggregations and coinciding with structural alterations of Purkinje cells (PC) in the brain, even prior to PC cell loss.
- (7) Our study takes the first real **translational approach by cross-validating the Nf changes found in humans with a SCA3 knock-in mouse model**, thus providing the basis for ongoing studies of NfL levels as outcome parameter for interventional trials with disease-modifying agents in SCA3, even for preclinical animal studies.
- (8) Finally, our study puts the Nf increase in SCA3 into a much broader, comprehensive differential biomarker perspective, as warranted for future clinical and preclinical trials in SCA3, by mapping a **biomarker cascade of SCA3**, comparing the evolution of NfL and pNfH with other biomarkers such as soluble ataxin-3 and aggregated ataxin-3 levels in the mouse model.

While we had already explicated the main differences of our present study compared to Wilke et al. 2018 and Li et al. 2019 in the initial version of our manuscript, we do appreciate the need to present both the commonalities and differences in more detail to allow the reader to rapidly appreciate the multiple qualitative novelties of our present study. We therefore added comparative paragraphs to the Introduction and Discussion sections.

3. Please include reference to previous study by Li et al. 2019 in the introduction (second paragraph, Page 4)

Reply: As recommended, we now included the reference to Li et al. 2019 already in the introduction.

4. Are the neuropathologic finding for SCA3 subject ID27925 somehow comparable with those of Li et al. 2019 (page 10)?

Reply: Our report is the first to include neuropathologic findings (both from humans and mice) in association with Nf levels. Our neuropathologic findings for SCA3 subject ID27925 suggest that NfL levels in SCA3 might not be related to major degeneration of corticospinal tract, primary motor cortex and Purkinje cell layer, but rather to degeneration of spinocerebellar tract, dentate nucleus and brainstem. In contrast, the report by Li et al. 2019 did not report any neuropathologic findings at all, but only standard volumetric MRI analyses. Given that these MRI analyses

by Li et al. 2019 were confined only to two predefined regions (only brainstem and cerebellar volumes) and did not include any advanced MRI approach which would have allowed e.g. quantifying fibre tracts, a comparison about tracts cannot be extrapolated from their study – particularly no differential hypothesis about relatively preserved vs. severely affected specific tracts and brain subregions, as presented in our report. This having said, our neuropathology-based hypothesis of an association of NfL levels to spinocerebellar tract and brainstem degeneration is at least broadly in line with the MRI-based findings by Li et al. 2019 which indicated an association of NfL levels to cerebellar volumes and brainstem volumes. At the same time, however, our findings provide a much more fine-grained anatomic hypothesis about specific brain tracts and regions, thus specifying these prior broad MRI associations. Following the reviewer's recommendation, we now included a comment on the consistency between our neuropathologic findings and the volumetric MRI findings by Li et al. 2019 in the Discussion section, but also a comment on the specification of these prior, rather broad associations through our novel findings.

5. Please include more information that compares this study with that of Li et. al, 2019 to the discussion (page 14).

Reply: We fully agree with the referee that more information comparing our study and the study of Li et al. 2019 is warranted. We now added an extensive paragraph to the Discussion section, including commonalities and differences between the two studies. For a detailed description of the novelty and additive value of our findings compared to Li et al. 2019, please see our reply to referee #2 point 2.

6. In the introduction on page 4, the authors write "We therefore assessed plasma NfL and pNfH also in a SCA3 knock-in mouse model (Martier et al.) (Haas et al., in preparation) across presymptomatic and symptomatic disease stages". Please complete reference Martier et al. (year and issue/page numbers) in the manuscript text and in the reference list. If the Haas et al. manuscript is still in preparation, it should be removed from the text and the correct and/or complete reference for the characterization of SCA3 knock-in mice added in replacement.

Reply: Following the reviewer's advice, we now completed the reference details for "Martier et al. 2019" in both the manuscript text and the reference list (Martier R, Sogorb-Gonzalez M, Stricker-Shaver J, Hubener-Schmid J, Keskin S, Klima J, et al. Development of an AAV-Based MicroRNA Gene Therapy to Treat Machado-Joseph Disease. *Mol Ther Methods Clin Dev* 2019; 15: 343-58.).

Moreover, we also included the complete reference to "Haas et al, in preparation" which is now available on the preprint server bioRxiv (Haas E, Incebacak RD, Hentrich T, Maringer Y,

Schmidt T, Zimmermann F, et al. A novel Ataxin-3 knock-in mouse model mimics the human SCA3 disease phenotype including neuropathological, behavioral, and transcriptional abnormalities. bioRxiv 2020: 2020.02.28.968024). Additionally, we now also provide more detailed information on the characterisation of the SCA3 knock-in mouse model in the current study, adding a section to the Appendix Supplementary Methods.

Referee #3

Comments on Novelty/Model System: The paper is well written and NfL is potentially a useful blood biomarker, although it has been used as a biomarker in other neurodegenerative disorders. Thus, the scientific impact is not high, especially regarding innovativeness. The scientific premise of the study is weak for SCA3 despite the knock-in mouse data. I am not sure whether the mouse model is a good model of human SCA3, and what they investigated was essentially histopathological correlation of their NfL data, verifying known histopathological studies of autopsy brains from human SCA3 patients. The mouse study should have added mechanistic understanding to the NfL elevation in blood, but none of these mechanisms were critically investigated. As for a clinical trial readiness, this study provides important information. Nevertheless, the paper will attract a lot of attention from pharmaceutical companies that are interested in this field for therapeutic development.

Remarks for Author: The value of this paper is that it provides encouraging data of a badly needed blood biomarker for clinical studies and drug trials for patients with SCA3 and pre-ataxia SCA3-mutation carriers. The use of two independent cohorts with consistent outcomes increases the confidence, and data from the animal model of SCA3 may assure the use of this serum biomarker in preclinical animal studies for the development of disease-modifying drugs for SCA3.

There are some weaknesses and concerns:

1. Blood NfL levels have attracted attentions of researchers as easily accessible biomarkers of various neurodegenerative disorders. Thus, the concept of using blood NfL levels as a biomarker in SCA3 is not particularly novel. Some authors of this paper have published a biomarker paper of NfL in MSA-C, SAOA and SCA1/2/3/6 earlier, and a separate group of investigators reported data on serum NfL levels in SCA3 subjects similar to this paper in November 2019 (Li QF, et al. Neurofilament light chain is a promising serum biomarker in spinocerebellar ataxia type 3. Mol Neurodegener. 2019 Nov 4;14(1):39).

Reply: *We appreciate the opportunity to explicate the novelty of our present study in comparison to (a) NfL increases in other neurodegenerative diseases and (b) previous research on NfL increases in SCA3. Before doing so, we wish to highlight that none of these studies (or any*

other study) has ever investigated **pNfH** for any repeat-expansion SCA, let alone for SCA3. Thus, apart from all the novelties related to NfL outlined below, the study of pNfH is completely new for SCA3. Thus, our study is setting the stage for this neurofilament as a novel biomarker likely applicable also to other repeat-expansion SCAs.

(a) Novelty in comparison to other neurodegenerative diseases

Indeed, as stated in the Introduction and Discussion sections of our initial manuscript, Nf blood levels are increased in many degenerative and non-degenerative neurological conditions, including several multi-systemic ataxias (although increases do not occur in all types of degenerative ataxia, see Wilke et al. 2018, indicating the need to evaluate Nf levels for each single type of degenerative ataxia specifically).

This fact, however, does not compromise the novelty of our findings, as blood Nf increases are not expected -and also not required- to serve as a **diagnostic** biomarker to differentiate SCA3 from other neurodegenerative diseases. Assessment of the ATXN3 repeat expansion (or the mutant ataxin-3 protein) already completely fulfils this biomarker purpose. Instead, to prepare upcoming treatment trials in SCA3 and other repeat-expansion SCAs, biomarkers are urgently needed for other purposes, specifically as **biomarker of disease severity, progression and subject stratification**, and potentially as trial outcome measure. A biomarker fulfilling these purposes, ideally cross-validated in an animal model for future preclinical trials, would be of high novelty for the whole degenerative ataxia field. Our study demonstrates for the first time that NfL blood levels could indeed provide such a biomarker in SCA3, by demonstrating that NfL levels might serve as -inter alia- biomarkers of disease severity, disease progression and subject stratification, with cross-validation in a SCA3 mouse model.

(b) Novelty in comparison to Wilke et al. 2018 and Li et al. 2019

Our findings do not only confirm the NfL increase in SCA3 and the association of NfL levels with cross-sectional disease severity described in previous research (Wilke et al. 2018, Li et al. 2019), but largely extend previous findings in multiple aspects (also compare reviewer #2 point 2). In brief, our study is the first to demonstrate, inter alia:

- the **positive association of NfL levels with individual longitudinal disease progression** in SCA3.
- the **high intra-individual analyte stability of Nf levels in first short-term longitudinal measurements** in SCA3, providing the basis for first **sample size estimations** for SCA3 treatment trials with Nf levels as outcome measure.
- the **increase of NfL levels in preataxic SCA3 subjects with temporal proximity to the estimated ataxia onset**, allowing to define a preconversion stage close to the onset of the ataxic stage which may serve as window of opportunity for future disease modifying therapies.

- **associations of Nf levels with neuropathological examinations**, including human neuropathology for hypothesis-generation regarding which neuronal systems may drive the Nf increase in SCA3, and including mouse neuropathology demonstrating that peripheral blood Nf changes are associated with earliest SCA3 brain processes, closely following ataxin-3 aggregations and coinciding with Purkinje cell (PC) structural alterations in the brain, even prior to PC cell loss.
- **Nf changes also in a SCA3 mouse model**, thus providing the translational basis for Nf levels as outcome parameter for interventional trials with disease-modifying treatments for preclinical animal studies.
- a **biomarker cascade of SCA3**, focussing not only on NfL and pNfH per se, but relating both biomarkers to a large biomarker cascade including also other molecular biomarkers such as soluble ataxin-3 and aggregated ataxin-3 levels in the mouse model.

While we had already explicated the main differences of our present study compared to Wilke et al. 2018 and Li et al. 2019 in the initial version of our manuscript by one long sentence, we now fully embrace the need to present the commonalities and differences in more detail, thus allowing the reader to rapidly appreciate the multiple qualitative novelties of our present study (also compare reviewer #2 point 2). We therefore now added an extensive comparative paragraph to the Discussion section highlighting the novelties of our study.

2. Neither NfL nor NfH is specific for SCA3. Not only other SCAs but also many other neurological diseases have shown elevated levels. Discussions need to include how the limited specificity should be addressed in clinical trial settings.

Reply: We fully agree with the reviewer. As already explicated in the Discussion section of our initial manuscript, Nf levels are indeed increased in many degenerative and non-degenerative neurological conditions, including several multisystemic ataxias (though NfL levels are not increased in all degenerative ataxias, see Wilke et al. 2018, J. Neurol.; moreover, pNfH has never been tested in any SCA prior to our current study). Thus, Nf blood levels provide a generic, easily accessible parameter for quantifying neuronal decay, both in SCA3 and in several other neurological diseases.

However, importantly, future treatment trials in SCA3 do not require fluid biomarkers (such as Nfs) as a **diagnostic** biomarker against other neurological diseases, as they will include subjects on the basis of a pathogenic ATXN3 repeat expansion – which is relatively easy and reliable to assess. Hence, the limited specificity of Nf levels as a diagnostic biomarker against other neurodegenerative diseases does not limit their utility for treatment trials in SCA3. Instead, such trials highly warrant fluid biomarkers for several other purposes. Specifically, treatment trials in SCA3 will require fluid biomarkers as **biomarkers of (1) disease severity, (2) disease progression, (3) disease stratification and (4) treatment response**.

Following the reviewer's advice, we now included all these aspects on how Nfs will be used in clinical trial settings in the Discussion section of the manuscript.

3. Discussions should also assess how this will be viewed in the Biomarker Qualification Program of FDA and equivalent standards of the EMA.

Reply: We thank the reviewer for this helpful hint – presenting the biomarker utility in terms of the Biomarker Qualification Program of the Food and Drug Administration (FDA) indeed helps to add further clarity to our manuscript! This qualification program requires the specification of the biomarker's context of use, defined by the biomarker category (using the FDA vocabulary: diagnostic, monitoring, predictive, prognostic, response, safety, and susceptibility/risk biomarker) and its intended use in drug development (see figure):

source: <https://www.fda.gov/media/102659/download>, 2020-03-06

According to this formalised regulatory framework, our data prepare the basis for the use of NfL serum levels in SCA3 as:

- (1) **monitoring biomarker:** to indirectly detect change in the degree of neuronal decay in both preataxic and ataxic mutation carriers, as evidenced by our findings.
- (2) **prognostic biomarker:** to stratify SCA3 subjects, e.g. to define a cohort of mutation carriers in the preconversion stage close to the onset of clinical SCA3 disease for inclusion in treatment trials, thus helping to define a window of opportunity in which future disease modifying therapies for SCA3 might be most effective, as evidenced by our findings.
- (3) **response biomarker:** to demonstrate the treatment-induced reduction of neuronal decay in SCA3 treatment trials, as being investigated by ongoing follow-up studies which directly build on our data.
- (4) **safety biomarker:** to indicate the presence of possible toxicity of a disease-modifying intervention which might cause additional, unintended neuronal decay (instead of the desired reduction thereof), as investigated by ongoing follow-up studies directly building on our data.

For the purpose of completeness, we would like to add that the following biomarker categories (again in FDA vocabulary) are **not** required from neurofilaments, as they are already fulfilled by testing the ATXN3 repeat-expansion:

(5) **diagnostic biomarker**: to establish the presence or absence of the disease-causing repeat expansion.

(6) **susceptibility/risk biomarker**: to identify individuals with a predisposition to develop SCA3.

Following the reviewer's advice, we now included a comment on the potential use(s) of NfL serum levels according to the biomarker qualification program in the Discussion section.

4. What biological events do NfL and NfH biomarkers represent? Are neuronal deaths or near deaths required to release neurofilaments? Is cell loss needed? OR is leaky plasma membrane or axonal degeneration sufficient to increase levels of neurofilaments? Authors determine the tissue concentration in the extract of homogenized cerebellum and frontal lobe tissues. However, is there a good correlation between blood and CSF and between CSF and the extracellular compartment in the neurofilament concentration? These scientific premises are important for the interpretation of the presented data. The introduction section should include what are known and unknown about the mechanism/process of NfL and NfH elevation in the blood.

Reply: We thank the reviewer for raising these important questions which, for reasons of brevity, we had addressed only briefly in the initial version of our manuscript. Following the reviewer's recommendation, we now address them in more detail, providing the following information already in the Introduction section of the article.

Release of Nfs. Under physiological conditions, low amounts of neurofilaments are constantly being released from neurons in an age-related manner, with higher Nf levels being observed in older ages (Khalil et al. 2020, Nat. Commun., Gaetani et al. 2019, JNNP; Disanto et al. 2017, Ann. Neurol.). The release of Nfs sharply increases in response to axonal damage irrespective of the underlying cause of axonal damage (e.g. traumatic, vascular, inflammatory, degenerative injury). Prolonged Nf release has been observed even after acute, non-sustained injury (e.g. traumatic brain injury) (Gaetani et al. 2019, JNNP; Khalil et al. 2018, Nat. Rev. Neurol.), possibly related to the long half-lives of Nfs (Paterson et al. 2019, Nat. Rev. Neurol.). It is yet unknown whether Nf elevations relate to passive release from damaged axons and/or upregulated protein production and secretion reflecting attempted axonal regeneration (Paterson et al. 2019).

Correlation between Nf levels in CSF and blood. While the correlation between Nf levels in the extracellular compartment and the cerebrospinal fluid (CSF) is also yet unknown, a close correlation between Nf levels in the CSF and the peripheral blood has been consistently established across a broad range of different neurodegenerative diseases by a large series of

studies (for meta-analyses and overviews, see Khalil et al. 2018, Nature Reviews Neurology; Gaetani et al. 2019, JNNP). This correlation has been confirmed for both NfL (Wilke et al. 2016, JNNP) and pNfH (Wilke et al. 2019, CCLM) also in our lab, and was recently also confirmed for SCA3 (Li et al. 2019, Mol. Neurodegener.). Within this correlation between blood and CSF, Nf levels in blood are considerably lower than levels in the CSF (NfL: 1.0-2.5%, pNfH: 2.5-10%) (Disanto et al. 2017, Ann. Neurol.), as also demonstrated previously by our lab for both NfL (Wilke et al. 2016, JNNP) and pNfH (Wilke et al. 2019, CCLM).

Biological/neurological events reflected by Nfs. In contrast to NfL, neurofilament heavy (NfH) undergoes considerable posttranslational modification (mainly by phosphorylation), which may be relevant for neurons to maintain axon calibre and thus conduction velocity (Poesen et al. 2018). While pNfH levels might correlate better with lower motor neuron involvement and NfL levels better with upper motor neuron involvement (Poesen et al. 2018, Front. Neurol.), tissue expression levels across brain regions do not show qualitative differences between NfL and NfH (GTEEx Portal, 2020).

Following the reviewer's advice, we now included in the Introduction section a paragraph describing what is currently known and unknown about the underlying mechanisms of Nf elevations in the blood, including the correlations between Nf levels in CSF and blood, as demonstrated and already reported also for SCA3.

5. It is disappointing that no data regarding #4 are collected in the mouse experiments. Immunofluorescent microscopy/CSF SIMOA to detect NfL/NfH may yield interesting insights. It is particularly valuable if the mouse model shows histological, immunofluorescent, and electron microscopic changes in the pre-ataxic stage.

Reply: Following the reviewer's advice, we now added a comprehensive immunohistochemical and histological assessment of the SCA3 mouse model to the manuscript, including comprehensive investigations of **multiple SCA3-vulnerable brain regions and coverage of multiple time points, including the preataxic stage.**

We assessed samples of cerebellar cortex, deep cerebellar nuclei, frontal cortex, pons and the pyramidal tract, comparing heterozygous 304Q SCA3 mice at 2, 6, 12 and 18 months of age to wildtype animals. Samples were assessed by (a) **immunohistochemistry** with ataxin-3 staining to capture the characteristic ataxin-3 aggregation, and (b) **histology with four different stainings (NfL, pNfH, Nissl, and Calbindin)** to capture numeric and structural aspects of SCA3 neurodegeneration.

- (a) **Immunohistochemistry** showed progressive ataxin-3 aggregation in regions typically affected by SCA3, particularly the cerebellum, pons and frontal cortex, beginning at 2-3 months and being definitely present at 6 months of age (Figure 9), thus preceding both the weight and the motor phenotype of the SCA3 mice, and partly preceding, partly paralleling the Nf increase.

(b) **Histology** demonstrated progressive loss of intact Purkinje cells (PCs) in the cerebellar cortex, starting as early as 2-3 months and becoming significant at 12 months of age (Figure 10, Appendix Figure S3). Interestingly, at the onset of the Nf increase in blood (i.e. month 6), **structural changes in PCs** were microscopically more prominent than absolute PC loss, which might suggest that **incipient structural alteration, and not only PC cell death, might contribute to Nf release** in SCA3. These PC alterations were accompanied by structural disturbance of the NfL and pNfH network in the cerebellar cortex (Figure 10). No marked overall atrophy of the deep cerebellar nuclei, pons, frontal cortex and pyramidal tract was observed (Figure EV5, Appendix Figure S4).

Thus, we leveraged our mouse model to elucidate the associated histological and immunohistochemical brain changes at the time of the Nf increase, all still at the presymptomatic stage: With statistically significant increases at month 6, peripheral blood Nf increases closely follow cerebellar ataxin-3 aggregations (starting at months 2-3, considerably increasing at month 6), parallel PC structural alterations, and even largely precede PC cell loss (start at 2-3 months, significant at month 12). In more general terms, peripheral Nf blood increases occur with strong effect sizes already at the earliest stages of SCA3 disease, shortly following the onset of SCA3 brain hallmarks (ataxin-3 aggregation) and coinciding with incipient SCA3 neurodegeneration (PC structural alterations) even prior to PC cell loss, and are not merely secondary to marked overall atrophy of SCA3-vulnerable brain regions.

These results from our novel immunohistochemistry and histology analyses in SCA3 mice across all disease stages were now added to the text throughout the manuscript (Figure 9, Figure 10, Figure EV5, Appendix Figures S3 and S4).

6. The lack of longitudinal data does not allow for calculation of the sensitivity to change (SRM).

Reply: Following the reviewer's recommendation, we now acquired longitudinal neurofilament data. There are two ways to calculating sample sizes for treatment trials based on longitudinal data, depending on the aim of the intervention: for reducing progression (sensitivity to change) or for reducing an abnormally increased level of a parameter to normal level (referencing the relative abnormality of the parameter to the intraindividual stability over time, for an analogous approach to Nf-based sample size calculations in neurodegenerative repeat-expansion disease, see Byrne et al. 2018, Science Translational Medicine).

While we indeed do not have long-term longitudinal data (which would be required to feed the calculation of sensitivity to change, but would likely require 1-2 years to obtain), **we have now acquired short-term longitudinal data** which allow assessing intraindividual analyte stability over time (quantifiable e.g. by intraclass correlation coefficients, ICCs).

We investigated Nf levels in SCA3 subjects with short-term longitudinal resampling (over 6 weeks), permitting the intraindividual stability of each analyte to be assessed (Figure 6). The

very high intraclass correlation (ICC) values of both NfL and pNfH revealed them to be highly stable, suggesting that intraindividual variation in the analytes is likely to be a minimal source of noise in natural history and treatment trials using Nf blood levels as outcome measure.

Based on these short-term longitudinal data, we are now able to deliver first **sample size estimations** for future treatment trials which aim to lower Nf blood levels in SCA3 (Figure 6).

These estimates show that ≈ 15 subjects per arm would suffice to detect therapeutic effects, even for therapeutic effect sizes as low as 20%. This number is considerably below the cohort sizes which would likely be required for clinical endpoints (e.g. SARA score) (Jacobi et al., 2015) and at least equal, if not lower than required for MRI endpoints (Reetz et al., 2013; Adanyeguh et al., 2018).

We now included the assessment of intraindividual analyte stability and the sample size estimations for both NfL and pNfH in the manuscript (Methods, Results, and Discussion, Figure 6).

7. How does NfL and NfH levels compare with the MR biomarkers in terms of sensitivity and specificity?

Reply: MRI studies in SCA3 demonstrated a specific neurodegenerative pattern, consisting of atrophy of brainstem, cerebellum and basal ganglia (Schulz et al. 2010, Stefanescu et al. 2015, Rezende et al. 2018, Peng et al. 2019) and white matter damage in cerebral and cerebellar peduncles (Rezende et al. 2018), starting before clinical onset and progressing from infratentorial structures up to the cerebral cortex (Rezende et al. 2018). Yet, even these most recent and advanced structural MRI studies do not report sensitivity and specificity for differentiating SCA3 subjects from controls. However, recent magnetic resonance spectroscopy (MRS) studies in SCA3 demonstrated decreased levels of the neuronal marker total N-acetylaspartate (NAA) and increased levels of the glial marker myoinositol (MI) in pons and cerebellum (Adanyeguh et al. 2015, Joers et al. 2018), allowing distinction of SCA3 subjects from controls with high accuracy (Adanyeguh et al. 2015: sensitivity 93%, specificity 100%; Joers et al. 2018: sensitivity 95%, specificity 100%). Our study demonstrates that blood levels of NfL are equally suitable to make this distinction between SCA3 subjects and controls (cohort #1: 99% sensitivity, 92% specificity; cohort #2: 93% sensitivity, 100% specificity), while pNfH achieves lower discriminatory performance (cohort #1: sensitivity 79%, specificity 73%; cohort #2: sensitivity 59%, specificity 96%). Given that the diagnosis of SCA3 is made by genetic test, however, MRI and fluid biomarkers should not primarily be interpreted as diagnostic biomarkers (as "sensitivity and specificity" might suggest), but rather as biomarkers for capturing neuronal damage and disease progression. Longitudinal analyses of volumetric MRI parameters demonstrated that volumes of the striatum, brainstem and cerebellum might be most sensitive to change in SCA3 (Reetz et al. 2013, Adanyeguh et al. 2018), outperforming clinical scores (such as the Scale for the Assessment and Rating of Ataxia, SARA) (Jacobi et al. 2015) to measure disease progression in SCA3 and potentially reducing the required sample size for treatment trials. With our sample size

estimates based on longitudinal samples (see reviewer #3 point 6), we demonstrate that Nf blood levels likewise allow reducing the sample size required for treatment trials, also outperforming established clinical scores. Thus, both MRI biomarkers and Nf levels, possibly even in combination, might be used to increase the power of upcoming treatment trials.

We now included a comparison of Nf levels with MR biomarkers in terms of sensitivity and specificity in the Discussion section. Moreover, we now discuss that Nf blood levels might allow reducing the sample size required for treatment trials, similar to the sample size reductions achievable by MR parameters.

8. What is the difference in utility between wt/mutant ATXN3 levels and NfL/NfH levels in clinical trials? Can NfL/NfH levels be useful as biomarkers of "target engagement" as authors imply?

Reply: *Given that our work is indeed the first study ever assessing all of these biomarkers (ataxin-3, NfL, pNfH) within one and the same study, we thank the reviewer to give us the opportunity to clarify their utility in clinical trials (hereby following the FDA framework of biomarker categories and contexts of use, also see our reply to reviewer #3 point 3).*

*While **ataxin-3 levels** might serve as **target engagement biomarkers** directly capturing the upstream effect of future molecular treatments on the key disease protein (ataxin-3), **NfL and pNfH levels** might serve as – more downstream – **biomarkers related to neuronal decay**. Particularly, NfL levels might serve as **monitoring biomarker** (for capturing neuronal decay), **prognostic biomarker** (e.g. to stratify preataxic SCA3 subjects in the preconversion stage), **response biomarker** (e.g. to demonstrate possible treatment-induced reduction of neuronal decay in SCA3 treatment trials) and/or **safety biomarker** (e.g. to indicate possible toxicity of disease-modifying interventions which might cause additional, unintended neuronal decay) (also see reply to point 2 by this reviewer #3). As NfL and pNfH capture the axonal decay effects downstream of the direct target (ataxin-3) engagement, they cannot serve as direct target engagement biomarkers.*

In fact, our study provides first preliminary support for the rationale that wt/mutant ataxin-3 levels might serve as target engagement biomarkers (but not as monitoring biomarkers for monitoring disease progression), while NfL/pNfH levels might serve as more downstream biomarkers related to neuronal decay, highlighting their differential utility. Specifically, while (both soluble and aggregated) ataxin-3 levels are the first biomarkers within the biomarker cascade investigated here to show alterations, thus indicating very early expression of the pathogenic key protein, they were of limited value as progression biomarker as they did not show any temporal evolution with disease progression and neuronal decay. In contrast, Nf levels – as generic biomarkers related to neuronal decay – rose slightly later than ataxin-3 levels, but still signalled the presence of neuronal decay already in presymptomatic animals.

Moreover, Nf levels showed a temporal evolution (and a correlation with disease severity in SCA3 subjects).

Thus, while mutant ataxin-3 levels might serve as target engagement biomarker for interventional trials aiming to reduce mutant ataxin-3 levels, Nf levels, particularly NfL, might be used to demonstrate a reduction of neuronal decay as the desired downstream effect of the treatment. Accordingly, ataxin-3 levels and Nf levels might possibly be combined as composite biomarkers to capture complementary treatment effects.

While we had already indicated different biomarker categories and contexts of ataxin-3 versus NfL/pNfH levels in the Results and Discussion sections of the initial version of our manuscript, we now realised that these passages might have been too short and not clear enough. We now explicate the differential utility of mutant ataxin-3 levels and Nf levels in the Discussion section, highlighting that both might be of complementary value to each other.

9. The repeat that was knocked in contained 304 trinucleotides in the (CAA CAG CAG)₄₈ configuration. The length of 304 trinucleotides is artificially long and may trigger different types of pathogenic mechanisms, such as those mediated by toxic gain of function by (CAG)_n expansion RNA and repeat-associated non-ATG (RAN) translation. Additionally, the presence of periodic CAA may alter the stability of the expanded repeat in the genomic DNA and binding to RNA-binding proteins by the RNA transcript. While such interrupted expanded repeats have been successfully used in other diseases, investigators of such models took painstaking verifications of the pathogenic authenticity.

Reply: We here decided to choose a knock-in (KI) SCA3 mouse model rather than one of the many existing SCA3 transgenic mouse models, as the latter express mutant ataxin-3 under artificial promoters, which would have confounded our current investigation of the biomarker evolution during the natural aging of the mice. However, the existing SCA3 KI mouse models – with polyQ expansions in the range of SCA3 patients – display only mild or no behavioural phenotypes (Ramani et al. 2015, Switonski et al. 2015, Ramani et al. 2017). This observation has also been made in KI models of other polyQ diseases, such as SCA1 (Lorenzetti et al. 2000), SCA2 (Damrath et al. 2012) and Huntington's disease (HD) (Menalled et al. 2003). Particularly, while we saw in rodent models of HD that it is not sufficient to introduce CAG expansions of the same repeat length as found in the pathogenic human HTT allele, we also learned that expansions of several folds of this length do allow triggering a phenotype in the rodent models (Farshim et al. 2018).

As we here aimed to select a mouse model which best captures the human phenotype under both molecular and behavioural aspects, we therefore chose a SCA3 KI mouse model with a hyper-expanded glutamine repeat. The 304Q KI mouse model chosen here indeed shows all main biochemical, behavioural, and transcriptomic features of human SCA3, and has indeed been verified for pathogenic authenticity on each of these levels (Haas et al. 2020). We now

added the reference to the characterisation and verification analyses, which have now been published (Haas et al. 2020, doi: <https://doi.org/10.1101/2020.02.28.968024>).

The 304Q SCA3 knock-in mouse model was indeed constructed to have an interrupted CAG-repeat (interruptions of the CAG-repeat by CAA, i.e. (CAGCAGCAA)_n) as (a) this allowed to ensure meiotic stability of the repeat length across generations (i.e. to keep the repeat length stable) (Frontali et al. 1999), and as (b) an interrupted repeat is also found in the murine ataxin-3 locus. Moreover, the interruptions also prevent interactions of the repeat RNA with the translation process (i.e. by prevention the formation of loops) and RAN-translation (repeat-associated non-AUG translation). On the protein level, both the interrupted (CAA-containing) repeat – as found in the murine ataxin-3 locus – and the uninterrupted (only CAG) repeat – as found in the human ataxin-3 locus – both lead to the same amino acid sequence required to model a polyglutamine disease. We now added these considerations to the manuscript (Appendix Supplementary Methods).

10. Motor phenotypes of the SCA3 mouse model should analyze males and females separately since the performance of mice may differ depending on the sex.

Reply: We agree that – as in many mouse models of neurodegenerative diseases (e.g. C9orf72) – there might be sex effects on the motor phenotype in SCA3 mouse models as well. Following the reviewer's recommendation, we now added – in addition to the overall group analysis – a separate sex-specific analysis. In this analysis, we confirmed the effect of a **motor phenotype** (quantified by the base of support of the hind paws) developing at age 18 months in the female group, which might be mainly driving the observed overall group effect of the motor phenotype. There was also a trend towards an effect in males, but in general larger group sizes per sex would be needed to draw more thorough conclusions on whether there is a sex-specific effect on the motor phenotype in this mouse model. This might indeed be an avenue for future work focussing on further detailed in-depth characterisations of the mouse model per se. As a first starting point for this, we now added the results of our sex-specific analysis to the supplement (Appendix Figure S2), as suggested by the reviewer.

Analogous to the sex-specific analysis of the motor phenotype, we also included **further sex-specific analyses for further aspects of our mouse model:**

Bodyweight. With both sexes pooled, heterozygous animals started differing from homozygous animals in weight at age 12 months. Analysing both sexes separately, we observed the onset of the weight phenotype for heterozygous animals at age 7 months in males and 12 months in females. Thus, both NfL and pNfH increases in SCA3 relative to wildtype mice started earlier (age: 6 months) than the weight phenotype (age: 7 months males, age: 12 months females) (Appendix Figure S1).

NfL. With both sexes pooled, heterozygous animals started differing from wildtype animals regarding NfL levels at age 6 months. Analysing both sexes separately, we confirmed the NfL

increase at age 6 months in females, and, though less pronounced, qualitatively also in males (Figure EV2).

pNfH. With both sexes pooled, heterozygous animals started differing from wildtype animals regarding pNfH levels at age 6 months. Analysing both sexes separately, we confirmed the pNfH increase at age 6 months in both females and males (Figure EV2).

In summary, both **NfL and pNfH increases** in heterozygous SCA3 mice relative to wildtype mice started earlier (age: 6 months, both males and females) (Figure EV2) than the **weight phenotype** (age: 7 months males, age: 12 months females) (Appendix Figure S1) and the **motor phenotype** (age: 18 months both males and females) (Appendix Figure S2). This confirmed and further highlights one of our main findings, namely the value of Nf levels as proximity blood biomarkers for the preconversion stage also in the SCA3 mouse model. Following the reviewer's advice, we now included the sex-specific analyses in the supplementary material (Figure EV3, Appendix Figures S1 and S2) and in the Discussion section.

11. How does the prediction of the ataxia onset in preataxic mutation carriers compare the prediction based on the CAG repeat size and the current age in the two Cohorts? (See Tezenas du Montcel et al. J Med Genet. 2014 Jul;51(7):479-86.)

Reply: For predicting the ataxia onset, we applied for all individual preataxic subjects – whether in cohort #1 (8 subjects) or in cohort #2 (14 subjects) – the same formula, namely exactly the formula by Tezenas du Montcel et al. 2014, to which the reviewer refers. That is, the prediction of ataxia onset was calculated in the same way in both cohorts. The preataxic subjects had a predicted time to ataxia onset of 9.1 years (7.7-13.4) (median and IQR) in cohort #1 and 9.8 years (8.8-14.3) in cohort #2. We now added this information to Table 1 of the revised manuscript.

The formula is based on the individual CAG repeat size and the current age. The values of the model are tabulated in Table 4 of the Supplement of the original article (Tezenas du Montcel et al. 2014. J. Med. Genet.). We now explicate the model in more detail in the Methods section.

We thank all referees for the very constructive feedback!

23rd Apr 2020

Dear Prof. Synofzik,

Thank you for the submission of your revised manuscript to EMBO Molecular Medicine. We have now received the enclosed reports from the referees that were asked to re-assess it. As you will see the reviewers are now supportive and I am pleased to inform you that we will be able to accept your manuscript pending the following final amendments:

1) Please provide a point-by-point letter INCLUDING my comments as well as the reviewer's reports and your detailed responses to their comments (as Word file).

2) Please carefully check the authors guidelines for formatting your supplemental information: Expanded view and Appendix (see:

<https://www.embopress.org/page/journal/17574684/authorguide#expandedview>)

- a scale bar in EV5 is missing

- move the method section in Appendix to the main article, along with references please, and change the call out to this page 14 of main article.

- Appendix information must be only present in the Appendix pdf file, delete from main article.

3) Figures: please simplify the number of figures. Quite a few of them have only 1 or 2 panels, it would be nice to reduce the overall number of figures down to 5 or 6.

4) Source Data: please zip together all files corresponding to figure EV5 and relabel source data according to the new labelling and organization of figures

5) In the main manuscript file, please do the following:

- correct/answer the track changes suggested by our data editors by working from the attached document

- in M&M, the statistical paragraph should reflect all information that you have filled in the Authors checklist, especially regarding randomisation, blinding, replication.

- indicate in legends exact $n=$ and exact $p=$ values, not a range, along with the statistical test used. Some people found that to keep the figures clear, providing an Appendix table Sx with all exact p -values was preferable. You are welcome to do this if you want to.

- Authors: those mentioned as members of a studygroup are not listed. Please note that the members of two other studygroups mentioned here were not listed by name. Perhaps it would be best to have all studygroup members listed together in the appendix?

- Callouts missing for Fig 7C-F and Fig EV5 is called out before Fig EV4.

6) Funding:

Please make sure to indicate in our submission system all sources of funding including grant numbers and to whom they are allocated.

7) Authors' contribution: please move the contribution of members of the studygroup to appendix

8) As part of the EMBO Publications transparent editorial process initiative (see our Editorial at <http://embomolmed.embopress.org/content/2/9/329>), EMBO Molecular Medicine will publish online a

Review Process File (RPF) to accompany accepted manuscripts.

In the event of acceptance, this file will be published in conjunction with your paper and will include the anonymous referee reports, your point-by-point response and all pertinent correspondence relating to the manuscript. Let us know whether you agree with the publication of the RPF and as here, if you want to remove or not any figures from it prior to publication.

9) Data availability section: please reformat this section and consider submitting the data to dbGAP or EGA which are specifically designed to respect patients confidentiality.

Please use the following format:

- [data type]: [full name of the resource] [accession number/identifier] ([doi or URL or identifiers.org/DATABASE:ACCESSION])

examples:

* RNA-Seq data: Gene Expression Omnibus GSExxxxx

(<https://www.ncbi.nlm.nih.gov/geo/query/acc.cgi?acc=GSExxxxx>)

* Chip-Seq data: Gene Expression Omnibus GSEyyyyy

(<https://www.ncbi.nlm.nih.gov/geo/query/acc.cgi?acc=GSEyyyyy>)

* patients' sequences: Database of Genotypes and Phenotypes (dbGAP) Xxxxxxx

(https://www.ncbi.nlm.nih.gov/projects/gap/cgi-bin/study.cgi?study_id=Xxxxxxx)

Please submit your revised manuscript within two weeks. I look forward to seeing a revised form of your manuscript as soon as possible.

I look forward to reading a new revised version of your manuscript as soon as possible.

Yours sincerely,

Celine Carret

Celine Carret, PhD

Senior Editor

EMBO Molecular Medicine

*** Instructions to submit your revised manuscript ***

To submit your manuscript, please follow this link:

(Link Not Available)

- 1) a .doc formatted version of the manuscript text (including Figure legends and tables)
- 2) Separate figure files*
- 3) supplemental information as Expanded View and/or Appendix. Please carefully check the authors guidelines for formatting Expanded view and Appendix figures and tables at <https://www.embopress.org/page/journal/17574684/authorguide#expandedview>
- 4) a letter INCLUDING the reviewer's reports and your detailed responses to their comments (as Word file).
- 5) The paper explained: EMBO Molecular Medicine articles are accompanied by a summary of the articles to emphasize the major findings in the paper and their medical implications for the non-specialist reader. Please provide a draft summary of your article highlighting
 - the medical issue you are addressing,
 - the results obtained and
 - their clinical impact.This may be edited to ensure that readers understand the significance and context of the research. Please refer to any of our published articles for an example.
- 6) For more information: There is space at the end of each article to list relevant web links for further consultation by our readers. Could you identify some relevant ones and provide such information as well? Some examples are patient associations, relevant databases, OMIM/proteins/genes links, author's websites, etc...
- 7) Author contributions: the contribution of every author must be detailed in a separate section.
- 8) EMBO Molecular Medicine now requires a complete author checklist (<https://www.embopress.org/page/journal/17574684/authorguide>) to be submitted with all revised manuscripts. Please use the checklist as guideline for the sort of information we need WITHIN the manuscript. The checklist should only be filled with page numbers where the information can be found. This is particularly important for animal reporting, antibody dilutions (missing) and exact values and n that should be indicated instead of a range.

9) Every published paper now includes a 'Synopsis' to further enhance discoverability. Synopses are displayed on the journal webpage and are freely accessible to all readers. They include a short stand first (maximum of 300 characters, including space) as well as 2-5 one sentence bullet points that summarise the paper. Please write the bullet points to summarise the key NEW findings. They should be designed to be complementary to the abstract - i.e. not repeat the same text. We encourage inclusion of key acronyms and quantitative information (maximum of 30 words / bullet point). Please use the passive voice. Please attach these in a separate file or send them by email, we will incorporate them accordingly.

You are also welcome to suggest a striking image or visual abstract to illustrate your article. If you do please provide a jpeg file 550 px-wide x 400-px high.

10) A Conflict of Interest statement should be provided in the main text

11) Please note that we now mandate that all corresponding authors list an ORCID digital identifier. This takes <90 seconds to complete. We encourage all authors to supply an ORCID identifier, which will be linked to their name for unambiguous name identification.

Currently, our records indicate that the ORCID for your account is 0000-0002-2280-7273.

(Link Not Available)

12) The system will prompt you to fill in your funding and payment information. This will allow Wiley to send you a quote for the article processing charge (APC) in case of acceptance. This quote takes into account any reduction or fee waivers that you may be eligible for. Authors do not need to pay any fees before their manuscript is accepted and transferred to our publisher.

Photos 400-800 DPI

*Additional important information regarding figures and illustrations can be found at <http://bit.ly/EMBOPressFigurePreparationGuideline>

The system will prompt you to fill in your funding and payment information. This will allow Wiley to send you a quote for the article processing charge (APC) in case of acceptance. This quote takes into account any reduction or fee waivers that you may be eligible for. Authors do not need to pay any fees before their manuscript is accepted and transferred to our publisher.

***** Reviewer's comments *****

Referee #2 (Remarks for Author):

The authors have added relevant information about the characterization of the animal model and fully addressed the previous comments and suggestions of this reviewer. The manuscript is significantly improved and will be of interest to a large number of scientists with interest in SCA3 and other neurodegenerative diseases.

Referee #3 (Comments on Novelty/Model System for Author):

Well performed study. All questions and comments that I have made have been addressed. There are no ethical issues.

Referee #3 (Remarks for Author):

This reviewer appreciates author's thorough and careful response and revisions.

Reply to comments by the Editor

1. Please provide a point-by-point letter INCLUDING my comments as well as the reviewer's reports and your detailed responses to their comments (as Word file).
2. Please carefully check the authors guidelines for formatting your supplemental information: Expanded view and Appendix
(see: <https://www.embopress.org/page/journal/17574684/authorguide#expandedview>).
 - a scale bar in EV5 is missing.
 - move the method section in Appendix to the main article, along with references please, and change the call out to this page 14 of main article.
 - Appendix information must be only present in the Appendix pdf file, delete from main article.

Reply: *We carefully checked the author guidelines for the formatting of the supplemental items and incorporated your corrections.*

- *We added the missing scale bar to EV5.*
- *We moved the text of the "Appendix Supplementary Methods" to the main text, along with the references, and removed the callouts accordingly.*
- *The updated Appendix is now only presented in the separate Appendix PDF file, not in the "clean version" of the manuscript. For transparency, the "track-changes version" of the manuscript still contains the Appendix.*

3. Figures: please simplify the number of figures. Quite a few of them have only 1 or 2 panels, it would be nice to reduce the overall number of figures down to 5 or 6.

Reply: *We managed to reduce the overall number of figures from 10 down to 7. We accordingly updated the figure captions, adjusted the figure callouts in the manuscript and relabelled the source data files.*

4. Source Data: please zip together all files corresponding to figure EV5 and relabel source data according to the new labelling and organization of figures.

Reply: *We zipped together the source data files which correspond to figure EV5. We relabelled the source data of all figures according to the new labelling of the figures.*

5. In the main manuscript file, please do the following:
 - correct/answer the track changes suggested by our data editors by working from the attached document.

Reply: We incorporated the suggested changes into the revised version of the manuscript by working from the attached document. Both the clean version and the track-changes version of our revised manuscript are based on the track-changes version provided by your data editors.

- in M&M, the statistical paragraph should reflect all information that you have filled in the Authors checklist, especially regarding randomisation, blinding, replication.

Reply: We ensured that all information provided in the author checklist is also included in the statistical paragraph of the Methods section (including randomisation, blinding and replication/validation).

- indicate in legends exact n= and exact p= values, not a range, along with the statistical test used. Some people found that to keep the figures clear, providing an Appendix table Sx with all exact p-values was preferable. You are welcome to do this if you want to.

Reply: We now reported the exact test statistics for each figure in Appendix Table S3, including all exact p-values, and ensured that the legends contain appropriate callouts to this Appendix item.

- AUTHORS: those mentioned as members of a studygroup are not listed. Please note that the members of two other studygroups mentioned here were not listed by name. Perhaps it would be best to have all studygroup members listed together in the appendix?

Reply: The author list, as presented on the first page of the article, now includes only one study group which summarises all study group members of our article. It is called the "SCA3 Neurofilament Study Group". As suggested, all study group members are now listed together in the appendix (Appendix Table S4).

- Callouts missing for Fig 7C-F and Fig EV5 is called out before Fig EV4.

Reply: We now included callouts to Fig 7C-F (old labelling). Given the reduction of the overall number of figures, these panels now correspond to Fig 5C-F (new labelling). We inserted a callout to Fig EV4 (in the Results section) before the callout to Fig EV5.

6. Funding: Please make sure to indicate in our submission system all sources of funding including grant numbers and to whom they are allocated.

Reply: We updated all funding sources in the submission system.

7. Authors' contribution: please move the contribution of members of the studygroup to appendix.

Reply: *The members of the “SCA3 Neurofilament Study Group” are now listed in Appendix Table S4, together with their affiliations. The section “Author contributions” only reported the contributions of the authors, but not the contributions of the study group members.*

8. As part of the EMBO Publications transparent editorial process initiative (see our Editorial at <http://embomolmed.embopress.org/content/2/9/329>), EMBO Molecular Medicine will publish online a Review Process File (RPF) to accompany accepted manuscripts. In the event of acceptance, this file will be published in conjunction with your paper and will include the anonymous referee reports, your point-by-point response and all pertinent correspondence relating to the manuscript.

Let us know whether you agree with the publication of the RPF and as here, if you want to remove or not any figures from it prior to publication. Please note that the Authors checklist will be published at the end of the RPF.

Reply: *We completely agree with the publication of the Review Process File (RPF).*

9. Data availability section: please reformat this section and consider submitting the data to dbGAP or EGA which are specifically designed to respect patients confidentiality.

Please use the following format:

- [data type]: [full name of the resource] [accession number/identifier] ([doi or URL or identifiers.org/DATABASE:ACCESSION])

Reply: *All source data of this study are now available in the supplementary material of the article. Thus, no additional external data repository outside of this article file package was necessary. We now re-phrased the data availability section accordingly.*

Reply to comments by the Referees

Referee #2

The authors have added relevant information about the characterization of the animal model and fully addressed the previous comments and suggestions of this reviewer. The manuscript is significantly improved and will be of interest to a large number of scientists with interest in SCA3 and other neurodegenerative diseases.

Reply: We thank the referee for the positive feedback.

Referee #3

Well performed study. All questions and comments that I have made have been addressed. There are no ethical issues. This reviewer appreciates author's thorough and careful response and revisions.

Reply: We highly appreciate the referee's positive feedback.

We thank all referees for the very constructive feedback!

10th May 2020

Dear Prof. Synofzik,

We are pleased to inform you that your manuscript is accepted for publication and is now being sent to our publisher to be included in the next available issue of EMBO Molecular Medicine.

Please read below for additional IMPORTANT information regarding your article, its publication and the production process.

Congratulations on your interesting work,

Celine Carret

Celine Carret, PhD
Senior Editor
EMBO Molecular Medicine

Follow us on Twitter @EmboMolMed
Sign up for eTOCs at embopress.org/alertsfeeds

*** ** IMPORTANT INFORMATION ** **

SPEED OF PUBLICATION

The journal aims for rapid publication of papers, using using the advance online publication "Early View" to expedite the process: A properly copy-edited and formatted version will be published as "Early View" after the proofs have been corrected. Please help the Editors and publisher avoid delays by providing e-mail address(es), telephone and fax numbers at which author(s) can be contacted.

Should you be planning a Press Release on your article, please get in contact with embomolmed@wiley.com as early as possible, in order to coordinate publication and release dates.

LICENSE AND PAYMENT:

All articles published in EMBO Molecular Medicine are fully open access: immediately and freely available to read, download and share.

EMBO Molecular Medicine charges an article processing charge (APC) to cover the publication costs. You, as the corresponding author for this manuscript, should have already received a quote with the article processing fee separately. Please let us know in case this quote has not been received.

Once your article is at Wiley for editorial production you will receive an email from Wiley's Author Services system, which will ask you to log in and will present you with the publication license form for completion. Within the same system the publication fee can be paid by credit card, an invoice, pro forma invoice or purchase order can be requested.

Payment of the publication charge and the signed Open Access Agreement form must be received before the article can be published online.

PROOFS

You will receive the proofs by e-mail approximately 2 weeks after all relevant files have been sent to our Production Office. Please return them within 48 hours and if there should be any problems, please contact the production office at embopressproduction@wiley.com.

Please inform us if there is likely to be any difficulty in reaching you at the above address at that time. Failure to meet our deadlines may result in a delay of publication.

All further communications concerning your paper proofs should quote reference number EMM-2019-11803-V3 and be directed to the production office at embopressproduction@wiley.com.

Thank you,

Celine Carret, PhD
Senior Editor
EMBO Molecular Medicine

Corresponding Author Name: Matthis Synofzik
Journal Submitted to: EMBO Molecular Medicine
Manuscript Number: EMM-2019-11803